# Attentional and executive functions in children and adolescents with developmental coordination disorder and the influence of comorbid disorders: A systematic review of the literature

**Catherine Lachambre[1,2], Mélodie Proteau-Lemieux[3], Jean-François Lepage[3], Eve-Line Bussières[4], Sarah Lippé[1,2]** *

1 Department of Psychology, Succursale Centre-Ville, University of Montreal, Montreal, Quebec, Canada,
2 Sainte-Justine Hospital Research Center, Montreal, Quebec, Canada, 3 Department of Pediatrics,
University of Sherbrooke, Sherbrooke, Quebec, Canada, 4 Department of Psychology, University of Quebec
at Trois-Rivières, Trois-Rivières, Quebec, Canada

* sarah.lippe@umontreal.ca

pone.0252043

(Spain), SPAIN

## Abstract

Developmental coordination disorder (DCD) is a neurodevelopmental disorder affecting primarily motor skills, but attentional and executive impairments are common in affected individuals. Moreover, the presence of neurodevelopmental comorbidities is frequent in this population, which certainly influences the cognitive profile of the children concerned. Previous studies have reported deficits in visuospatial/nonverbal and planning tasks. This systematic review of the literature aims to determine if impairments can be found in other attentional and executive functions as well. The type of cognitive tasks, the tasks' modality (verbal/nonverbal), and the influence of comorbid disorders on attentional and executive profiles are systematically considered. Forty-one studies were identified through the PubMed/Medline and PsycINFO databases according to pre-established eligibility criteria. The results reveal weaknesses in inhibitory control, working memory, planning, nonverbal fluency, and general executive functioning in children with DCD. The presence of comorbid disorders seemingly contributes to the verbal working memory difficulties findings. This review contributes to a better understanding of the cognitive impairments in DCD and of the needs of children with this disorder, allowing to optimize practitioners' therapeutic interventions.

## 1. Introduction

### 1.1. Developmental coordination disorder

Developmental coordination disorder (DCD) is a neurodevelopmental disorder that affects
motor skills development in children and influences their ability to perform multiple activities

**Data Availability Statement:** All relevant data are within the manuscript and its Supporting Information files.

**Funding:** The author(s) received no specific funding for this work.

**Competing interests:** The authors have declared that no competing interests exist.

of daily living [1, 2]. According to the Diagnostic and Statistical Manual of Mental Disorders–Fifth Edition [3], DCD is characterized by difficulties in movement acquisition and execution. Diagnostic criteria include motor abilities that are significantly inferior to those expected given the individual chronological age and learning opportunities, and impairments significantly interfering with daily activities, academic and/or professional accomplishments and hobbies, which emerge during a child's early development and are not better explained by intellectual disability, neurological condition affecting movement or visual impairment. Motor difficulties can be manifested as clumsiness, slowness and inaccuracy in movement execution.

Significant signs of DCD usually emerge during school years, but motor difficulties associated with DCD are developmental and tend to persist into adolescence [4] and adulthood [5]. Its prevalence is estimated between 5 and 6% in children aged 5 to 11 years old [3]. More boys than girls are affected, with reported ratios of men to women ranging from 1.9:1 to 7.3:1 [6, 7]. Great variability exists within countries, with prevalence among school-age children ranging from 1.8% in the United-Kingdom [7], 8% in Canada and 19% in Greece [8]. DCD is present across all cultural, ethnic and socioeconomic groups, but daily living activities and functional impacts of the disorder may vary within these different groups. Thus, these variables should be considered at the time of diagnosis.

DCD has been formerly referred to as *clumsy child syndrome*, and today the terms *specific developmental disorder of motor function* and *dyspraxia* can also be used to describe DCD [3, 9]. The terms DCD and dyspraxia (or developmental dyspraxia) are the most frequently used terms in the literature. DCD focusses on the observable symptoms, on their functional impacts and manifestations in daily living, and its diagnosis often belongs to occupational therapists or to physiotherapists [10, 11]. As for dyspraxia, the term is used interchangeably with DCD, and is widely used and diagnosed in neuropsychology [10–13]. Dyspraxia is often defined as difficulties with organizing, planning, executing and coordinating movement, which leads to impairments in acquisition of complex movements and of movement sequences [10, 12, 14]. It relies on a developmental conception of the brain, implying a motor cognition disorder that includes visual and spatial processing problems [11]. Children with dyspraxia and DCD typically show visuoperceptual, visuospatial, visuoconstructives and visuomotor deficits [12, 15, 16], further impacting school achievement and daily living.

Several theoretical models of dyspraxia have been proposed over the years to provide a coherent view of its core elements. The one proposed by Vaivre-Douret and her team [13] offers a thorough definition as it stipulates that dyspraxia involves a deficit of both execution of voluntary gesture and planning/programming of movement. According to this model, when a child intends to perform an action, he first has to formulate a plan to execute that movement, that considers perceptual information, then to build a mental representation of the movement by coding spatiotemporal parameters, and finally to execute the movement while correcting it according to sensorimotor feedback [13, 14]. By conducting neuropsychological, neuro-psychomotor and neuro-visual assessments in children with dyspraxia, the authors showed that the planning and programming processes were the core problems in this disorder, and that the execution mechanisms were disturbed only when dyspraxia was comorbid with other neuropsychological disorders [13]. Therefore, the question arises as to whether children with dyspraxia (or DCD) present only planning impairments on motor tasks, or broader planning and executive functioning deficits.

### 1.2. Neurodevelopmental comorbidities

It seems that children who only have DCD are the exception rather than the rule, comorbid disorders being present in most of the cases [17–19]. Attention deficit/hyperactivity disorder

(ADHD) is the most frequent comorbid condition, occurring approximately in 50% of the cases [3, 4, 6, 20, 21]. Besides, some authors report that DCD and ADHD overlap in their symptoms, as children with ADHD frequently demonstrate motor difficulties [22], and that executive dysfunctions and slow processing speed, which are typical features of ADHD, are often present in children with DCD [20]. It is thus unclear which difficulties are inherent to which disorder, but ADHD and DCD must still be considered as separate disorders since their core deficits are distinct [23]. Learning disorders, including developmental dyslexia (DDL) [24, 25], specific language impairment (SLI) [17, 26, 27], behavioral problems [17, 28] and autism spectrum disorder (ASD) [6, 29] are also common in children with DCD, concomitance between these disorders and DCD reaching up to 30 to 50%. Since deficits in executive and attentional functioning are generally part of these disorders [30–32], it seems essential to consider their presence and potential influence on executive and attentional capacities in children with DCD.

## 1.3. Executive and attentional functions

Executive functions are generally conceptualized as a set of general high order control processes [33] working together [34] to direct and manage cognitive, emotional and behavioral functions, especially during active problem solving [35]. A variety of cognitive abilities falls under the umbrella of executive functions, including working memory, inhibition, cognitive flexibility/shifting, goal-setting, planning, organization, self-regulation and fluency [33–37]. To this day, however, it is still unclear exactly how many components define executive functions and under which terms they should be grouped [38]. Indeed, several models of executive functions have been proposed, integrating different components and establishing interactions between them. One of the most integrative and complete models was proposed by Diamond [36]. According to this model, working memory is defined as the ability to hold information in mind for a short period of time while mentally manipulating it to execute a task, and it includes verbal and visuospatial subcomponents. Inhibitory control is described as the ability to control attention, behavior, thoughts and emotions to override a dominant, automatic or prepotent response. The author also explains that inhibitory control is divided into two subcomponents: interference control, combining cognitive and attentional inhibition, and response inhibition, defined as behavioral inhibition. It also encompasses self-regulation, which includes attentional and response inhibition, while focusing primarily on emotional control and regulation. According to Diamond [36], working memory and inhibitory control support each other: working memory allows one to maintain their goals in their mind along with what they should or should not do, and inhibitory control allows one to stay focused on the important working memory content by inhibiting distractors. When working together, these two executive functions allow cognitive flexibility, defined as one's ability to see things from another perspective and to shift between tasks. According to this model, fluency skills are part of cognitive flexibility, since one must be able to shift between different mind sets to be fluent in generating various ideas. Higher-level executive functions are also underpinned by working memory, inhibitory control and cognitive flexibility in this model: reasoning and problem-solving, which are synonymous with fluid intelligence, and planning skills [36].

Attentional functions can be divided into three central components [39, 40]: alertness, selective attention, and divided attention. Alertness allows individuals to adopt and sustain a state of vigilance and is typically involved in long, monotonous tasks. It contrasts with the second component, selective attention, which is the ability to selectively process a specific kind of information while ignoring distractors and involves filtering mechanisms. The third subfunction of attention is divided attention and concerns the notion of sharing mental processing

capacities, which appears necessary when performing two or more tasks simultaneously [39, 40]. These attentional functions are also closely related to executive functions, particularly to inhibitory control and working memory; inhibition capacities allow attentional orienting and control [36], while the allocation of attentional resources to specific information stream will dictate the ones accessing working memory [41].

Executive dysfunction leads to resistance to change, incapacity to modify non-optimal behaviors, impulsivity and risk-taking, social difficulties, avolition and learning difficulties [34, 36, 42]. For this reason, executive functions are essential to children's optimal daily functioning and their future quality of life, in their affective, social as well as academic spheres [11, 36].

## 1.4. State of knowledge

Several studies have reported impairments in executive and attentional functions in children with DCD [21, 43–45]. While some authors report a generalized executive dysfunction in this population [20], others report deficits in specific domains of attentional or executive functioning [45, 46], which can be more severe in nonverbal modalities [47]. Consequently, and according to the theoretical model of dyspraxia previously discussed, the question arises as to whether the impairments are only present in visuospatial/nonverbal modalities and on planning tasks, and thus are more linked to the primary deficits found in DCD, or whether difficulties can be found across a broader range of executive functions. This systematic review of the literature aims to answer this question, while considering the influence of comorbid neurodevelopmental disorders combined with DCD on attentional and executive profiles found in these children. This review is an important step in the acquisition and gathering of supplemental evidence regarding the attentional and executive deficits found in DCD. Indeed, consolidating the results found in previous research appears relevant, especially given that several studies based their conclusions on small samples.

On a clinical level, the purpose of this review is to shed light on impairments that can be a part of DCD rather than being explained by the comorbid disorders often observed in this disorder, and to better identify the needs of children with DCD considering their potentially complex clinical picture, in order to optimize their assessment and the interventions carried out. This systematic review is the first to focus exclusively on the cognitive profile associated with DCD, and more specifically on the attentional and executive functions in young individuals with this disorder, while explicitly considering the possible influence of co-occurring disorders.

## 2. Method

This systematic review was performed based on the guidance outlined in the Preferred Reporting Items for Systematic Reviews and Meta-Analysis (PRISMA) [48].

### 2.1. Search strategy and selection criteria

A systematic literature research was performed in PubMed/Medline and PsycINFO databases, including articles published between January 1980 and May 2020. Considering that pediatrics neuropsychology was poorly developed before 1980, it was chosen as the earliest publication date. Due to the multiple ways of conceptualizing attentional and executive functions in terms of their components, and to be as inclusive as possible, our research was made using all terms previously mentioned that are used to describe these functions. Thereby, research was conducted in English with the following keywords: (1) "developmental coordination disorder" OR "dyspraxi*" OR "motor skills disorder" OR "specific developmental disorder of motor function" OR "clumsiness" OR "clumsy child syndrome"; (2) AND "executive function*" OR "goal

setting" OR "set-shifting" OR "shifting" OR "switching" OR "flexibility" OR "planning" OR "inhibit*" OR "working memory" OR "organis*" OR "organiz*" OR "self-regulation" OR "fluency" OR "attention*"; (3) AND "child*" OR "adolescen*" OR "teen*" OR "youth" OR "schoolchild*" OR "preschool*". Publications referenced in the included articles were also screened to find additional articles.

Studies were included if (1) their participants were children or adolescents (17 years of age or younger; studies including older participants were excluded), (2) they had a group of participants with a diagnosis of DCD made by a health care professional, by a score at or below the 5th percentile on the *Movement Assessment Battery for Children* (MABC first or second edition), as this score indicates a significant movement difficulty [49], or by DSM (IV, IV-TR or 5) criteria for DCD combined with a movement ability measure, in which case a total score at or below the 15th percentile on the MABC was accepted, (3) their participants did not explicitly have any medical condition that could affect their motor or cognitive abilities, (4) they measured one or more attentional or executive functions using performance tests, (5) they used normative data of standardized measures or a control group comprising healthy individuals for results comparison, (6) they were published in French or English in a peer-reviewed journal and (7) they had an empirical research design.

Studies were first selected according to their title. Second, abstracts were read by two authors (CL and MPL) and studies that did not meet the eligibility criteria listed above were excluded. Then the same two authors screened full articles independently to ensure that all eligibility criteria were met. Their disagreements were discussed to reach a consensus and, whenever necessary, another author (SL) settled. The remaining articles were entirely read by the first author.

## 2.2. Data extraction

Data extraction from articles that met the selection criteria was made by the two first authors. Information was organized in an spreadsheet and included: title of the article, authors, year of publication, journal in which it was published, aims of the study, groups, their origin and samples size, gender and age of the subjects assessed (mean, standard deviation and range, when available), country in which the study took place, inclusion and exclusion criteria, presence of comorbid disorders in the sample and their nature, information about assessment of motor functions and confounding variables, cognitive functions assessed and tasks used, statistical analysis, results, limitations and commentaries about the paper. Subsequently, the relevant information was analyzed and summarized in Table 1: the authors and year of publication, the sample size, the gender of the participants and the number of males, the mean age of the participants, the presence of comorbid disorders, the attentional or executive function studied, the task(s) used to evaluate the function, and a summary of the results. Components of attentional and/or executive functioning assessed in each included study were determined according to what the study purported to measure. When the same score on a task was reported to assess more than one component of attentional or executive functioning, or to measure different components in different studies, results were reported only in relation to the component it was the most associated with, according to the *Compendium of Neuropsychological Tests* [50] or the task's reference. When several scores were available on the same task and associated with different components of attentional or executive functioning, information provided by each score was considered as a measure of its respective component.

## 2.3. Quality assessment of included studies

Quality assessment of studies included in this review was conducted using a checklist we developed based on the Newcastle-Ottawa Quality Assessment Scale (NOS) [51]. The NOS consists

**Table 1. Studies included in the systematic review.**

| Studies Authors (year) | Groups (sample size) Origin | Gender Number of males (%) | Age (years) M (SD) | Presence of a comorbid disorder to DCD | Attentional or executive function studied | Task(s) used to assess attentional or executive functions | Summary of results |
|---|---|---|---|---|---|---|---|
| Alesi et al. (2019) | DCD (18) TD (18) *Schools* | 9 (50.0%) 9 (50.0%) | 4.6 (0.9) 4.6 (0.9) | Exclusion of children with intellectual disability, visual or neurological impairment and neurodevelopment-tal disorders. | Fluency (verbal) Response inhibition (verbal) Working memory (verbal) Working memory (visuospatial) | BVN-5-11 Stroop Task, Gift Wrap Task, Snack Delay Task Working Memory Tasks: Forward word recall, Selective word recall, Verbal dual task Pathway recall, Selective pathway recall, Visuospatial dual task | Children with DCD performed significantly more poorly than TD children on both the forward ($p = .022$) and selective word recalls ($p = .05$), but the results did not differ significantly for the verbal dual task. They also had lower scores on the three visuo-spatial tasks ($p < .05$). Children with DCD also performed more poorly than TD children on the BVN-5-11($p = .00$). Children with DCD obtained lower scores for both congruent and incongruent stimuli on the Stroop task ($p < .01$). However, there was no significant difference between DCD and TD children on emotionally-valenced tasks |
| Alloway (2007) | DCD (55; same sample as Alloway, 2011 and Alloway et al., 2009) *Schools, referred by health care professio-nals* | 44 (80.0%) | 8.8 (1.6) | No mention of exclusion. | Working memory (verbal) Working memory (visuospatial) | AWMA: Listening Recall, Counting Recall, Backwards Digit Recall Odd-One-Out, Mr. X, Spatial Span | 49% of sample obtained standard scores of less than 85 on the verbal WM measures. 60% of sample obtained scores of less than 85 on the visuospatial WM measures. The difference between performance on visuospatial and verbal WM measures was not significant. |
| Alloway (2011) | DCD (55; same sample as Alloway, 2007 and Alloway et al., 2009) ADHD (50) TD (50) *Schools, referred by health care professio-nals* | 44 (80.0%) 43 (86.0%) 30 (60.0%) | 8.8 (1.6) 9.8 (1.0) 9.9 (1.0) | Exclusion of children diagnosed with behavioral or attentional problems. | Working memory (verbal) Working memory (visuospatial) | AWMA: Listening Recall, Counting Recall, Backwards Digit Recall Odd-One-Out, Mr. X, Spatial Span | The DCD and ADHD groups performed more poorly than the TD group on both verbal and visuospatial WM measures ($p < .05$). Children with DCD showed deficits on both modalities compared to TD controls. Their WM profile did not differ significantly compared to that of children with ADHD. |
| Alloway & Archibald (2008) | DCD only (11) DCD unselected (12) SLI (11) *Schools, referred by health care professio-nals* | 8 (72.7%) 8 (66.7%) 7 (63.6%) | 8.9 (1.4) 8.5 (1.6) 8.8 (1.4) | Exclusion of children diagnosed with ADHD or ASD. Inclusion of children with language or nonverbal reasoning difficulties (DCD unselected group). | Working memory (verbal) Working memory (visuospatial) | AWMA: Listening Recall, Counting Recall, Backwards Digit Recall Odd-One-Out, Mr. X, Spatial Span | The two DCD groups did not significantly differ on their WM profiles. Their performance was more than 1.25 SD below the standardized mean on most tasks. Children in the DCD only group performed significantly worse than children with SLI on the visuospatial WM measures, even when the contribution of receptive language skills was accounted for. Their verbal WM skills were similar. Thus, children in DCD groups had deficits in both WM modalities, whereas those in SLI group only had verbal WM deficits. |

*(Continued)*

**Table 1.** (Continued)

| Studies Authors (year) | Groups (sample size) Origin | Gender Number of males (%) | Age (years) M (SD) | Presence of a comorbid disorder to DCD | Attentional or executive function studied | Task(s) used to assess attentional or executive functions | Summary of results |
|---|---|---|---|---|---|---|---|
| Alloway et al. (2009) | DCD (55; same sample as Alloway, 2007, 2011)<br>SLI (15)<br>ADHD (83)<br>AS (10)<br>*Schools, referred by health care professio-nals* | 44 (80.0%)<br><br>9 (60.0%)<br>71 (85.5%)<br>8 (80.0%) | 8.8 (1.6)<br><br>9.2 (1.7)<br>9.1 (1.1)<br>8.8 (1.5) | Exclusion of children diagnosed with behavioral problems. | Working memory (verbal)<br>Working memory (visuospatial) | AWMA:<br>Listening Recall, Counting Recall, Backwards Digit Recall<br>Odd-One-Out, Mr. X, Spatial Span | Children with DCD had significantly lower scores than children with AS on visuospatial WM measures ($p < .05$), but the difference was attributable to the motor component of the tests. There were no significant differences when comparing with the other clinical groups. On verbal WM measures, there were no differences between groups. |
| Alloway & Temple (2007) | DCD (20)<br>MLD (20)<br>*Schools, referred by health care professio-nals* | 14 (70.0%)<br>15 (75.0%) | 9.8 (1.4)<br>9.8 (1.4) | No mention of exclusion. | Working memory (verbal)<br>Working memory (visuospatial) | AWMA:<br>Listening Recall, Counting Recall, Backwards Digit Recall<br>Odd-One-Out, Mr. X, Spatial Span | Mean standard scores of children with DCD on all WM tasks were less than 85. Performances of children with DCD were poorer than that of children with MLD on all WM measures ($p < .01$). Children with DCD showed significant deficits in visuospatial WM, while MLD group performed within age-expected level in this domain. On measures of verbal WM, both groups were impaired. Thus, impaired visuospatial WM seems to be more specific to children with DCD when compared to children with MLD. |
| Asonitou & Koutsouki (2016) | DCD (54)<br>TD (54)<br>*General population* | 36 (66.7%)<br>36 (66.7%) | Total of 42 5-year-olds and 66 6-year-olds: 5.5 (0.4) | Exclusion of children with any other medical or neurological condition, or identified intellectual disability. | Selective attention (visual)<br><br>Planning | CAS:<br>Expressive Attention, Number Detection, Receptive Attention<br>Matching Numbers, Planned Codes, Planned Connections | A significant proportion of children with DCD were impaired on planning and attention measures ($p < .01$). Globally, children with DCD had more difficulties than TD children in all measured domains, and difficulties in planning were greater than those in attention. |
| Asonitou et al. (2012) | DCD (54)<br>5-year-old (24)<br>6-year-old (30)<br>TD (54)<br>5-year-old (18)<br>6-year-old (36)<br>*General population* | 36 (66.7%)<br>13 (54.2%)<br>23 (76.7%)<br>37 (68.5%)<br>15 (83.3%)<br>22 (61.1%) | 42 5-year-olds: 5.2 (0.3)<br>66 6-year-olds: 5.8 (0.2) | Exclusion of children diagnosed with emotional or behavioral disorder, with a history of pre- or existing developmental disorder (such as ADHD), or with an intellectual disability (IQ < 70). | Selective attention (visual)<br><br>Planning | CAS:<br>Expressive Attention, Number Detection, Receptive Attention<br>Matching Numbers, Planned Codes, Planned Connections | Children with DCD performed more poorly than TD children on all tasks ($p < .05$), indicating difficulties in terms of both attention and planning when compared to healthy controls. Also, more severe motor impairment seems to be associated with poorer planning abilities. |
| Barray et al. (2008) | DD (32)<br>AP (16)<br>*Learning disorders centers* | 24 (75.0%)<br>6 (37.5%) | 9.1 (2.3)<br>9.0 (2.1) | Exclusion of children with probable personality disorder, social conduct disorder, or verbal IQ < 80. | Selective attention (visual)<br>Selective attention (auditive)<br>Planning<br>Fluency (nonverbal) | NEPSY:<br>Visual Attention<br><br>Auditory Attention and Response Set<br><br>Tower task<br>Design Fluency | Children with DD performed significantly better than those with AP only on the Visual attention task ($p = .022$). All children had difficulties in the Design Fluency task that seemed to be the most difficult task for them. Performances on the other tasks were globally within the normative range. |

*(Continued)*

**Table 1.** (Continued)

| Studies Authors (year) | Groups (sample size) Origin | Gender Number of males (%) | Age (years) M (SD) | Presence of a comorbid disorder to DCD | Attentional or executive function studied | Task(s) used to assess attentional or executive functions | Summary of results |
|---|---|---|---|---|---|---|---|
| Bernardi et al. (2017) | DCD (17) MD (17) TD (17) *General population* | 6 (35.3%) 8 (47.1%) 13 (76.5%) | 12.0 (1.2) 10.5 (0.6) 11.3 (1.0) | Exclusion of children diagnosed with ADHD, ASD, reading, language and IQ < -2 SD, or any medical condition. | Working memory (verbal) Working memory (nonverbal) Fluency (verbal) Fluency (nonverbal) Response inhibition (verbal) Response inhibition (nonverbal) Planning (verbal) Planning (nonverbal) Flexibility (verbal) Flexibility (nonverbal) | WMTBC: Listening Recall Odd-One-Out test D-KEFS: Verbal Fluency D-KEFS: Design Fluency Verbal VIMI (total errors) Motor VIMI (total errors) D-KEFS: Sorting test (verbal sorts) D-KEFS: Sorting test (nonverbal sorts) D-KEFS: Trail Making test CANTAB: Intra-/Extra-Dimensional Shift | Children with DCD performed significantly more poorly than TD children on all nonverbal measures ($p < .001$) and on the verbal fluency task ($p = .001$) at both time points (2-year follow-up). Children with MD also had poorer performance on all nonverbal measures, except the flexibility one. At time 2, only nonverbal WM ($p = .005$) and nonverbal fluency ($p = .047$) differences remained between MD and TD groups. Improvement over time was significant for verbal ($p < .001$) and nonverbal ($p = .002$) WM, fluency and flexibility($p < .001$), and for nonverbal planning ($p = .013$). It was not significant for verbal and nonverbal inhibition and verbal planning. Changes over time were similar between each group. |
| Bernardi et al. (2016) | DCD (23) MD (30) TD (38) (Same sample as Leonard et al., 2015) *General population* | 16 (69.6%) 17 (56.7%) 17 (44.7%) | 10.0 (1.1) 8.9 (1.2) 9.3 (1.0) | Exclusion of children diagnosed with any other neurodevelopment-tal disorder, including ADHD. | Response inhibition (verbal) Response inhibition (nonverbal) | Verbal VIMI (total errors and total completion time) Motor VIMI (total errors and total completion time) | On the motor inhibition task, children with DCD or MD had difficulties performing accurately. On the verbal inhibition task, they took significantly more time than TD children ($p = .002$), but the accuracy between groups was similar. Inhibition impairments in DCD and MD groups thus appear to affect accuracy when a motor response is required and completion time when the response is given verbally. There were no significant differences between DCD and MD groups. |
| Biotteau et al. (2017) | DCD (22) DCD+DDL (23) DDL (20) *Referred by a learning disability center or health care profession-nals* | 16 (72.7%) 16 (69.6%) 12 (60.0%) | 9.7 (1.6) 9.9 (1.2) 10.2 (1.3) | Exclusion of children with intellectual disability, SLI or ADHD, according to DSM-IV-TR criteria. Presence of DDL, excluding surface dyslexia, in DCD+DDL group. | Sustained attention (visual) Response inhibition (nonverbal) Working memory (verbal) | CPT-II (omission errors) CPT-II (commission errors) WISC-IV: Digit Span and Letter-Number Sequencing | All children had mean scores within or slightly above the normal range on the CPT-II. Variations between groups were not significant. WM index was within the normal range in the three groups. Having a dual diagnosis did not lead to a cumulative impact on cognitive abilities. |
| Blais et al. (2017) | DCD (10) TD (10) *Hospital* | 7 (70.0%) 3 (30.0%) | 13.5 (1.4) 13.5 (1.8) | Exclusion of children with a history of head trauma or epilepsy, an intellectual disability or ADHD according to DSM-5 criteria. | Sustained attention (visual) Response inhibition (nonverbal) | CPT-II (omission errors) CPT-II (commission errors) | Children with DCD performed significantly worse than TD children. Their average percentages of commission, omission and perseveration errors were significantly greater than those of control individuals ($p < .05$). There was no difference between groups for reaction times. |

*(Continued)*

**Table 1.** (Continued)

| Studies Authors (year) | Groups (sample size) Origin | Gender Number of males (%) | Age (years) M (SD) | Presence of a comorbid disorder to DCD | Attentional or executive function studied | Task(s) used to assess attentional or executive functions | Summary of results |
|---|---|---|---|---|---|---|---|
| Chen et al. (2012) | SDCD (20) MDCD (46) TD (36) *Research database* | 9 (45.0%) 16 (34.8%) 24 (66.7%) | 9.5 (0.3) 9.6 (0.3) 9.7 (0.3) | Exclusion of children with any signs of neurolo-gical of physical impairments, developmental disorders or intellectual disability, accor-ding to a rehabi-litation physician. | Attentional inhibition (endogenous mode of orienting attention) | COVAT | The capacity of children with DCD, both severe and moderate, to intentionally disengage their attention from invalid cued location when the delay between precue and target stimulus was longer was reduced. This suggests a deficit of attentional control in DCD, more specific to the endogenous mode of orienting attention and referring to the process of disengagement inhibition. |
| de Castelnau et al. (2007) | DCD (24) TD (60) *Hospital* | 18 (75.0%) 30 (50.0%) | 3 age groups: 8–9 years, 10–11 years and 12–13 years. 8 children with DCD and 20 TD children in each of them. | Exclusion of children with ADHD, according to a brief neuropsychological examination and DSM-IV criteria, and with an IQ < 80. | Sustained attention (visual) Response inhibition (nonverbal) | CPT double version (correct responses) CPT double version (commission errors) | Children with DCD had significantly less correct responses than TD children ($p < .0001$), but it increased with age ($p < .05$). They omitted significantly more responses than controls. However, there was no significant difference in number of commission errors between groups, and these decreased with age. In conclusion, children with DCD had poorer attentional capacities than TD children, but they were not more impulsive, and their capacities improved with age. |
| Dyck & Piek (2010) | DCD (20) RELD (21) PLA (22) PMC (28) *Referred by health care profession-nals* | 13 (65.0%) 16 (76.2%) 14 (63.6%) 17 (60.7%) | 8.4 (2.1) 7.1 (1.8) 7.5 (1.6) 8.8 (3.1) | Exclusion of children diagnosed with any other comorbid disorders. | Response inhibition (nonverbal) Working memory (verbal) | Go/No Go task (commission errors) Trailmaking/Me-mory Updating task, Goal Neglect task | Children with DCD had scores within the normal range on the response inhibition task. Their performances were within the low normal range for WM tasks. Differences with children with PMC were not significant. Scores of children in the RELD group were lower on the response inhibition and WM tasks ($p < .05$), indicating greater difficulties in this group. |
| Gonzalez et al. (2016) | DCD (10) TD (12) *Specialized clinic* | 7 (70.0%) 8 (66.7%) | 10.1 (1.0) 10.0 (1.1) | Exclusion of children diagnosed with ADHD. | Attentional inhibition (exogenous mode of orienting attention) | Visuospatial attention task cued and non-cued conditions | Children with DCD made more inhibition errors (saccades to the cue) than TD children ($p = .002$), indicating poor inhibitory control. These inhibition difficulties resulted in inappropriate allocation of visual attention. |
| Kaiser & Albaret (2016) | DCD (7) ADHD (9) TD (15) *Specialized school, referred by health care profession-nals* | Not mentioned | 9.9 (0.9) 10.2 (1.3) 9.6 (1.3) | Exclusion of children with a dual diagnosis of DCD and ADHD, and with an IQ < 70. | Selective attention (visual) Alertness (visual) Divided attention | KITAP: Distractibility Alerting Divided Attention | Results of children with DCD on KITAP tasks were not significantly different from results of children with ADHD or of TD children. Thus, the three groups cannot be discriminated based on their performance on the KITAP. |

*(Continued)*

**Table 1.** (Continued)

| Studies Authors (year) | Groups (sample size) *Origin* | Gender Number of males (%) | Age (years) M (SD) | Presence of a comorbid disorder to DCD | Attentional or executive function studied | Task(s) used to assess attentional or executive functions | Summary of results |
|---|---|---|---|---|---|---|---|
| Kirby et al. (2010) | DCD (11) TD (28) *Research database* | 8 (72.7%) 13 (46.4%) | 9.9 10.2 (SDs not available) | Exclusion of children with severe learning difficulties or low cognitive ability, according to teachers. Inclusion of children with ADHD symptoms. | Planning | River Crossing task | Significant differences between groups were seen in children's planning strategies. Children with DCD tended to add mats (p < .05), but not modify their placement of mats, whereas TD children were more likely to change spacing of mats. This suggests that initial mat placement plan of TD children was more effective than that of children with DCD. This indicates that children with DCD may have difficulty to elaborate a plan as efficiently as TD children and may also not know how to improve it. |
| Leonard et al. (2015) | DCD (23) MD (30) TD (38) (Same sample as Bernardi et al., 2016) *General population* | 16 (69.6%) 17 (56.7%) 17 (44.7%) | 10.0 (1.1) 8.9 (1.2) 9.3 (1.0) | Exclusion of children diagnosed with ASD or ADHD, and of children with IQ, language or reading skills more than two SD below the mean. | Working memory (verbal) Working memory (nonverbal) Fluency (verbal) Fluency (nonverbal) Response inhibition (verbal) Response inhibition (nonverbal) Planning (verbal) Planning (nonverbal) Flexibility (verbal) Flexibility (nonverbal) | WMTBC: Listening Recall Odd-One-Out test D-KEFS: Verbal Fluency D-KEFS: Design Fluency Verbal VIMI (total errors) Motor VIMI (total errors) D-KEFS: Sorting test (verbal sorts) D-KEFS: Sorting test (nonverbal sorts) D-KEFS: Trail Making test CANTAB: Intra-/Extra-Dimensional Shift | For nonverbal WM, fluency and inhibition tasks, children with MD or DCD scored significantly lower than TD children (p < .001). For nonverbal planning, only the MD group (not the DCD group) differed significantly from the TD group (p < .001). There were no significant differences between groups on flexibility tasks or on any of the verbal measures. In summary, children with DCD or MD had more difficulties on nonverbal tasks compared to TD children. |
| Mandich et al. (2002) | DCD younger + older (20) TD younger + older (20) *Specialized clinic* | Not mentioned | Younger: 8.6 (0.8) Older: 10.6 (0.7) Younger: 8.5 (0.7) Older: 10.8 (0.8) | Exclusion of children diagnosed with ADD. | Response inhibition (nonverbal) Attentional inhibition (exogenous mode of orienting attention) | Visual Simon task (failure-to-inhibit errors) Visual Simon task (reaction times) | Children with DCD seem to have no deficit regarding the time needed to suppress an incorrect response and this ability appears to develop at the same rate in both groups. However, children with DCD had a reliably smaller rate of correct responses in the incompatible trials compared to TD children (p = .011). Overall, children with DCD exhibited an inhibitory dysfunction regarding manual response inhibition, manifesting itself in terms of error rate. |

*(Continued)*

**Table 1.** (Continued)

| Studies Authors (year) | Groups (sample size) Origin | Gender Number of males (%) | Age (years) M (SD) | Presence of a comorbid disorder to DCD | Attentional or executive function studied | Task(s) used to assess attentional or executive functions | Summary of results |
|---|---|---|---|---|---|---|---|
| Mandich et al. (2003) | DCD (18) TD (18) *Specialized clinic* | Not mentioned | 9.9 (1.5) 9.8 (1.4) | No mention of exclusion. | Response inhibition (nonverbal) Attentional inhibition (endogenous mode of orienting attention) | Go/No Go task with both informative and uninformative precue conditions (anticipation and failure-to-inhibit errors) Go/No Go task with both informative and uninformative precue conditions (reaction times) | Children with DCD did not commit more anticipation errors than TD children, but they produced about twice as many failure-to-inhibit errors (p = .007). Children with DCD may have impaired inhibitory control, since they needed more time to intentionally disengage their attention from the cued position (disengagement inhibition). They also exhibited difficulty to inhibit the unwanted movement of attention urged by precues. Thus, they exhibited more difficulties than controls in restraining both their manual and attentional movements. |
| Piek et al. (2007) | DCD (18) ADHD-I (20) ADHD-C (19) TD (138) *Referred by health care profession-nals* | 12 (66.7%) 16 (80.0%) 15 (78.9%) 59 (42.8%) | 8.8 (2.0) 10.8 (1.8) 10.7 (2.3) 10.3 (2.2) | Exclusion of children with an estimated IQ < 80. | Working memory (verbal) Flexibility | Trailmaking/Memory Updating task, Goal Neglect task Visual Inspection Time task | On the Goal Neglect task, children with DCD made more errors than the three other groups (p < .05). On the Trailmaking/Memory Updating task, children in the DCD group were significantly slower than those in other groups (p < .01) but did not commit more errors. They were also slower than the other groups on the set-shifting task. Thus, performance of children with DCD was significantly poorer than that of ADHD and TD children on all measures of EF, and there was no significant difference between ADHD and TD children. |
| Pratt et al. (2014) | DCD (26) TD (24) *General population* | 22 (84.6%) 13 (54.2%) | 9.9 (2.5) 9.6 (2.0) | Exclusion of children with any other disorder, such as ADHD, ASD or dyslexia. | Planning (reduced motor-load) Planning (high motor-load) Response inhibition (reduced motor-load/verbal) Response inhibition (high motor-load/nonverbal) | NEPSY: Tower task Rotational Bar task Stroop task NEPSY: Knock-Tap task | Children with DCD performed significantly more poorly than TD children on both the high motor-load and reduced motor-load planning tasks. They also had significantly lower scores than TD children on the reduced motor-load inhibition task, but differences in performance on the high motor-load inhibition task were not significant. Results might have been influenced by the tasks' complexity. |
| Querne et al. (2008) | DCD (9) TD (10) *Hospital* | 7 (77.8%) 7 (70.0%) | 9.9 (1.8) 10.0 (1.1) | Exclusion of children with a history of neurological or psychiatric disorders. | Sustained attention (visual) Response inhibition (nonverbal) | Go/No Go task (omission errors) Go/No Go task (commission errors) | Children with DCD did not make more commission errors than TD children, but they made significantly more omission errors (p = .011). Thus, children with DCD seem to be as effective as TD children in inhibiting a prepotent motor response, but to have more difficulties with sustained attention. |

(*Continued*)

**Table 1.** (Continued)

| Studies Authors (year) | Groups (sample size) Origin | Gender Number of males (%) | Age (years) M (SD) | Presence of a comorbid disorder to DCD | Attentional or executive function studied | Task(s) used to assess attentional or executive functions | Summary of results |
|---|---|---|---|---|---|---|---|
| Rahimi-Golkhan-dan et al. (2016) | DCD (12)<br>TD (24)<br>*Schools, general population* | 4 (33.3%)<br>10 (41.7%) | 9.8 (1.4)<br>10.3 (1.6) | Exclusion of children diagnosed with intellectual disability, or neurological or psychiatric disorders such as ADHD. | Sustained attention (visual)<br><br>Response inhibition (nonverbal) | Go/No Go task with positively- and negatively-valenced stimuli (omission errors)<br>Go/No Go task with positively- and negatively-valenced stimuli (commission errors) | Children with DCD were more impulsive than TD children with happy faces stimuli, as they made more commission errors ($p = .017$). The difference between groups for sad faces was not significant. There was no significant difference between the two groups for omission errors. Overall, results showed a deficit of inhibitory control in the DCD group when the no-go stimulus was a compelling, positively-valenced cue, which shows a deficit in 'hot' executive functions. |
| Ruddock et al. (2016) | DCD (62)<br>6 years (7)<br>7 (13)<br>8 (11)<br>9 (11)<br>10 (15)<br>11 (5)<br>TD (109)<br>6 years (28)<br>7 (21)<br>8 (21)<br>9 (18)<br>10 (15)<br>11 (5)<br>12 (1)<br>*Schools, general population* | 35 (56.5%)<br>3 (42.8%)<br>9 (69.2%)<br>9 (81.8%)<br>7 (63.6%)<br>6 (40.0%)<br>1 (20.0%)<br>48 (44.0%)<br>9 (32.1%)<br>7 (33.3%)<br>10 (47.6%)<br>12 (66.7%)<br>7 (46.7%)<br>2 (40.0%)<br>1 (100.0%) | 6.4 (0.6)<br>7.5 (0.3)<br>8.6 (0.4)<br>9.5 (0.3)<br>10.6 (0.3)<br>11.3 (0.2)<br>6.4 (0.4)<br>7.5 (0.3)<br>8.4 (0.3)<br>9.5 (0.3)<br>10.3 (0.3)<br>11.4 (0.2)<br>12.3 (0.0) | Exclusion of children with a history of developmental (ASD, ADHD), physical and/or neurological condition reported in a parent/teacher pre-screening questionnaire. | Response inhibition (nonverbal) | Double-Jump Reaching task–Modified version | Until 10–11 years old, children with DCD performed slower and with more variability than TD children. Therefore, in general, they were less efficient than controls, but the developmental lag in their performance seemed to lessen in later childhood.<br>Performance of both groups improved with age, but the growth curve was different. |
| Ruddock et al. (2015) | DCD (42)<br>6–7 years (10)<br>8–9 (16)<br>10–12 (16)<br>TD (87)<br>6–7 years (26)<br>8–9 (38)<br>10–12 (23)<br>*Schools, general population* | 22 (52.4%)<br>5 (50.0%)<br>11 (68.8%)<br>6 (37.5%)<br>34 (39.1%)<br>9 (34.6%)<br>15 (39.5%)<br>10 (43.5%) | 7.3 (0.7)<br>8.9 (0.6)<br>11.1 (0.4)<br>7.2 (0.5)<br>8.9 (0.6)<br>10.7 (0.5) | Exclusion of children for whom any developmental, neurological and/or physical condition was reported, which was confirmed by the child's school health officer. | Response inhibition (nonverbal) | Double-Jump Reaching task–Modified version | Children with DCD were generally slower than TD children ($p = .002$) and they made significantly more anticipation errors, indicating difficulties in inhibitory control. Moreover, only the differences between younger children and the two other groups were significant ($p = .005$); mid-aged and older children did not significantly differ in their anticipation error rate. Thereby, initiation inhibition capacities are poorer in children with DCD than in TD children, but the gap between them lessen when they get older. |

*(Continued)*

**Table 1.** (Continued)

| Studies Authors (year) | Groups (sample size) Origin | Gender Number of males (%) | Age (years) M (SD) | Presence of a comorbid disorder to DCD | Attentional or executive function studied | Task(s) used to assess attentional or executive functions | Summary of results |
|---|---|---|---|---|---|---|---|
| Sartori et al. (2020) | DCD (63) MD (31) TD (63) *Schools* | 39 (62.0%) 20 (65.0%) 39 (62.0%) | 8.7 (0.64) 8.9 (0.74) 8.7 (0.63) | Exclusion of children with a neurological condition or with ADHD. | Flexibility Response inhibition (nonverbal) Response inhibition (verbal) Working memory (verbal) Working memory (visuospatial) | Five Digits Test, Trail Making Test Go/No Go app (motor response) Go/No Go app (verbal response) Hayling Test, Oral Word Span in Sentences Odd-One-Out | DCD and MD children were slower on the Oral Word Span and the Odd-One-Out than TD children ($p < .001$). Children with DCD performed more poorly than TD children on both tasks (verbal and motor responses) of the Go/No Go app ($p < .005$) and on the Hayling test ($p < .001$). DCD children made more mistakes and had longer completion times on parts 2, 3 and 4 of the Five Digits Test ($p < .001$), and MD children made more mistakes than TD children on parts 3 and 4 of the Five Digit Test ($p = .01$). Children with DCD also performed more poorly than TD children on both parts of the Trail Making Test. No statistical differences were found between DCD and MD children. |
| Sumner, Pratt, & Hill (2016) | DCD (52) TD (52) *General population* | 36 (69.2%) 36 (69.2%) | 9.2 (2.2) 9.3 (1.5) | Exclusion of children diagnosed with any other developmental disorder (ADHD, ASD, dyslexia), neurological condition, or other medical condition that could explain motor impairment. | Working memory (verbal) | WISC-IV: Digit Span, Letter-Number Sequencing | Scores of children with DCD were significantly lower than TD children on the WM index ($p = .02$). Children with DCD performed significantly worse than TD children on the Digit span task ($p < .05$), but not on the Letter-number sequencing task. Almost 30% of children with DCD had scores of more than 1 SD below the population mean on the Digit span. Thus, the results showed some indication of difficulties in WM. |
| Thornton et al. (2018) | DCD (9) DCD+ ADHD (18) ADHD (20) TD (20) *Schools, advertising in hospitals and physicians' offices* | 6 (67.0%) 15 (83.0%) 18 (90.0%) 8 (40.0%) | 13.6 (2.74) 10.9 (2.62) 12.4 (2.8) 10.6 (1.67) | Exclusion of children with a visual or intellectual impairment and a neurological or medical disorder. | Response inhibition (nonverbal) | Go/No Go Task (commission errors) | Children with co-occurring DCD and ADHD made more commission and total errors than TD children ($p < .05$). However, the reaction times and omission errors did not differ between groups. |

*(Continued)*

**Table 1.** (Continued)

| Studies Authors (year) | Groups (sample size) Origin | Gender Number of males (%) | Age (years) M (SD) | Presence of a comorbid disorder to DCD | Attentional or executive function studied | Task(s) used to assess attentional or executive functions | Summary of results |
|---|---|---|---|---|---|---|---|
| Toussaint-Thorin et al. (2013) | DD (13) TD (14) *Hospital* | 11 (84.6%) 8 (57.1%) | 10.1 (1.4) 10.4 (1.3) | Exclusion of children with possible associated dyslexia and with a verbal IQ < 70. Inclusion of children with ADHD. | Selective attention (auditive) Flexibility Planning Response inhibition (nonverbal) Fluency (verbal) General executive functioning | NEPSY: Auditory Attention and Response Set Trail Making test A and B for Children NEPSY: Tower task, BADS-C: 6-Part test The Paired Images test NEPSY: Verbal Fluency Ecological assessment: cooking task | No deficits were found on measures of attention, flexibility and fluency. Almost half of children with DD had a score in the clinical range on the inhibition task. Three children with DD had a pathological score on the tower task, but planning appeared to be more impaired on the 6-Part test, the strategy score being particularly affected. Thus, children with DD exhibited deficits of nonverbal planning and inhibition ($p < .05$). On the cooking task, children with DD made significantly more errors than TD children ($p < .001$) and they were not able to inhibit their verbalizing behavior, despite the task's guidelines. They were also significantly more dependent than TD children ($p = .034$), suggesting a limited capacity to find strategies or solutions to face a problem. |
| Tsai et al. (2012) | DCD (24) TD (30) *General population* | 12 (50.0%) 15 (50.0%) | 11.6 (0.3) 11.7 (0.4) | Exclusion of children with ADHD according to DSM-IV criteria, with any known neurological disorder, behavioral problems, pervasive developmental disorders or special educational needs, or an IQ < 85 or > 115. | Alertness (visual) Working memory (visuospatial) | Visuospatial Working Memory Paradigm (non-delay condition) Visuospatial Working Memory Paradigm (delay condition) | Children with DCD performed less accurately than TD children in the conditions with delays ($p < .001$), but not in the non-delay condition. This indicates difficulty in WM tasks, but not in the attentional task. EEG results showed that children with DCD allocated fewer resources to compare spatial locations ($p < .001$). Overall, a deficit in retrieval of spatial information was found in children with DCD through EEG measures, which seemed to impair their WM performance. |
| Tsai et al. (2010) | DCD (30) TD (30) *General population* | 15 (50.0%) 15 (50.0%) | 9.5 (0.3) 9.6 (0.2) | Exclusion of children with ADHD according to DSM-IV, any defi-nite signs of neuro-logical disorders, behavioral pro-blems or pervasive development disor-ders, special needs in education, or an IQ < 85 or > 125. | Attentional inhibition (exogenous mode of orienting attention) | Visuospatial attention (eye-gaze cueing) paradigm | There were no significant differences between groups on error rates. Children with DCD responded significantly slower than TD children in all conditions ($p < .001$). This suggests a reduced alertness to the imminent appearance of the target. Results showed a deficit in reflexive/ automatic (exogenous) orienting of visual attention. |

*(Continued)*

**Table 1.** (Continued)

| Studies Authors (year) | Groups (sample size) Origin | Gender Number of males (%) | Age (years) M (SD) | Presence of a comorbid disorder to DCD | Attentional or executive function studied | Task(s) used to assess attentional or executive functions | Summary of results |
|---|---|---|---|---|---|---|---|
| [1]Tsai et al. (2009) | DCD (28) TD (26) *General population* | 12 (42.9%) 12 (46.2%) | 9.5 (0.3) 9.5 (0.3) | Exclusion of children with ADHD according to DSM-IV, any definite signs of neurological disorders or behavioral problems, special needs in education, or an IQ < 85 or > 125. | Attentional inhibition (endogenous mode of orienting attention) | Endogenous Posner Paradigm | Children with DCD responded more slowly than TD children (*p < .0001*), but there was no significant difference on the error rate. They had more difficulties than TD controls to move their attention when it has been primed to a falsely indicated location. Thus, they exhibited an inhibition deficit in the endogenous mode of orienting attention. |
| [2]Tsai et al. (2009) | DCD-LEs (36) TD (36) *General population* | 19 (52.8%) 19 (52.8%) | 9.9 (0.5) 9.8 (0.5) | Exclusion of children with ADHD according to DSM-IV, special educational needs, physical or behavioral problems, or evident neurological damage. | Attentional inhibition (endogenous mode of orienting attention) Attentional inhibition (exogenous mode of orienting attention) | COVAT  Visual Simon task (reaction times) | Children with DCD took longer than TD children to respond to neutral trials, indicating a reduced alertness to the imminent appearance of the target. Results showed a deficit in DCD children associated with only the intentional disengagement of attention (endogenous mode; *p < .001*) but not the automatic/reflexive dislocation of attention (exogenous mode). |
| Wang et al. (2015) | DCD (23) TD (23) *General population* | 12 (52.2%) 12 (52.2%) | 9.4 (0.5) 9.3 (0.5) | Exclusion of children with ADHD according to DSM-IV criteria, with any definite signs of neuro-gical disorders, behavioral problems or special educational needs, or an IQ < 85 or > 125. | Attentional inhibition (exogenous mode of orienting attention) | Visuospatial attention (eye-gaze cueing) paradigm | There were no significant differences between groups on error rates. Children with DCD responded significantly slower than TD children in all conditions (*p < .001*). They had a poorer general attentional orienting capacity and a difficulty to effectively alter a planned action after their attention had been incorrectly oriented. |
| Williams et al. (2013) | DCD (10) DCD+ADHD (16) ADHD (14) TD (18) *Schools, hospital or referred by health care professio-nals* | 6 (60.0%) 14 (87.5%) 8 (57.1%) 10 (55.6%) | 8.5 (1.2) 9.1 (1.7) 10.1 (1.4) 10.2 (1.3) | Exclusion of children with ADHD or DCD in other groups than the ones they are supposed to be in, with any physical or neurological condition that could contribute to motor impairment, or with an IQ < 70. | Sustained attention (auditive) | TEA-Ch: Score! | Children in the DCD+ADHD group had significantly more difficulty in the sustained attention task compared to the TD children. Scores in the DCD group were also lower than in the ADHD group, but the differences were not significant. There were no other differences between groups. |
| Wilson & Maruff (1999) | DCD (20) TD (20) *General population* | 10 (50.0%) 10 (50.0%) | 10.3 10.6 | Exclusion of children with current or history of neurological diseases, including head injury, psychiatric disorders, including ADHD, or an estimated IQ < 80. | Attentional inhibition (endogenous and exogenous modes of orienting attention) | COVAT | Within each group, responses were significantly faster for valid than invalid cues, but the effect was much greater for children with DCD (*p < .001*). Even when the delay between the cue and the target was longer, these children still had difficulty to shift their attention as efficiently as TD children. Results showed a deficit in the endogenous disengagement of attention in children with DCD, while the exogenous orienting mode was not affected. |

*(Continued)*

**Table 1.** (Continued)

| Studies Authors (year) | Groups (sample size) Origin | Gender Number of males (%) | Age (years) M (SD) | Presence of a comorbid disorder to DCD | Attentional or executive function studied | Task(s) used to assess attentional or executive functions | Summary of results |
|---|---|---|---|---|---|---|---|
| Wilson et al. (1997) | DCD (20) TD (20) *General population* | 18 (90.0%) 18 (90.0%) | 9.8 9.7 | Exclusion of children with current or past history of neurological diseases, including head injury, psychiatric disorders, including ADHD, or as estimated IQ < 80. | Attentional inhibition (endogenous and exogenous modes of orienting attention) | COVAT | Children with DCD responded slower than TD children ($p = .001$) and increasing delay between the cue and the stimulus did not help them as it did for TD children. They were able to complete attentional orienting as efficiently as TD children when the delay was brief. Reaction time was also significantly greater in the DCD group for invalid cues, compared with the TD group ($p = .001$). These results showed a deficit in the disengagement of attention in children with DCD. Thus, they have a deficit in the endogenous mode of orienting attention, but not in the exogenous mode. |
| Zhu et al. (2012) | DCD (39) TD (39) *Hospital* | 28 (71.8%) 28 (71.8%) | 8.1 (0.5) 8.0 (0.7) | Exclusion of children with any comorbid neurological diseases, mental and neurodevelop-mental disorders, according to detailed physical and psychiatric examinations. | General executive functioning | Wisconsin Card Sorting Test | Children with DCD committed significantly more errors, perseverative responses and perseverative errors than TD children ($p < .001$). They also needed more trials to complete the first category ($p < .001$). Results indicated difficulties in executive functioning in children with DCD. |

Note: M = mean; SD = standard deviation; DCD = developmental coordination disorder; TD = typically developing; BVN-5-11 = Development Neurological Assessment;
AWMA = Automated Working Memory Assessment; WM = working memory; ADHD = attention-deficit/hyperactivity disorder (-I = impulsive; -C = combined); SLI = specific language
impairment; ASD = autism spectrum disorder; AS = Asperger syndrome; IQ = intellectual quotient; MLD = mild learning difficulties; CAS = Das-Naglieri Cognitive Assessment System;
DD = developmental dyspraxia; AP = dyspraxia following preterm birth; NEPSY = Developmental Neuropsychological Assessment; MD = motor difficulties (without DCD); WMTBC = Working
Memory Test Battery for Children; D-KEFS = Delis-Kaplan Executive Function System; VIMI = Verbal Inhibition Motor Inhibition test; CANTAB = Cambridge Neuropsychological Test
Automated Battery; DDL = developmental dyslexia; CPT-II = Continuous Performance Test, Second Edition; WISC-IV = Wechsler Intelligence Scale for Children, Fourth Edition; SDCD = severe
developmental coordination disorder; MDCD = moderate developmental coordination disorder; COVAT = Covert Orienting of Visuospatial Attention Task; RELD = mixed receptive expressive
language disorder; PLA = relatively poor language ability; PMC = relatively poor motor coordination; KITAP = computerized test battery of attention for children; MAND = McCarron Assessment
of Neuromuscular Development; BADS-C = Behavioural Assessment of the Dysexecutive Syndrome in Children; BOT-2-SF = Short Form Bruininks-Oseretsky Test of Motor Proficiency, second
edition; DCD-LEs = DCD on lower extremities; TEA-Ch = Test of Everyday Attention for Children.

hree domains: "selection of subjects", "comparability of subjects" and "outcome", each domain including two to three items. Although the use of the NOS has been established in meta-analyses [52], it was first developed for case-control and cohort studies. Therefore, its use was not suitable for the present study. Since we could not find a tool giving standardized criteria for assessing the quality of neuropsychological and behavioral studies, we adapted the NOS based on the methods used by Wu et al. [53–55] and Caçola et al. [56], that were inspired by the NOS and the PRISMA standards. We developed three to four quality items for each domain of the NOS (selection, comparability and outcome). The result is a 10-item checklist, including: inclusion/exclusion criteria and samples source (for the selection domain), comparability of samples regarding age, gender and IQ (for the comparability domain), and description of outcome measures, adequacy of outcome analysis and discussion (for the outcome domain; see Appendix 1 for the complete checklist). Items could be answered by "yes" or "no", and quality level of evidence was rated as high (8 "yes" or more), medium (6–7 "yes") or low (5 "yes" or less). The quality assessment was carried out by the first two authors independently and any discrepancy was discussed until they reached a decision by consensus.

## 3. Results

### 3.1. Characteristics of included studies

A total of 1088 articles were identified through databases; 708 remained after removing duplicates. Of these, twenty articles (2.8%) had to be discussed between two authors to reach a consensus regarding their inclusion or exclusion, and a third had to settle for five (0.7%) of them. As a result, 41 studies were included in this systematic literature review (see flow diagram in Fig 1), representing 1,097 children and adolescents with DCD (or developmental dyspraxia; DD) with or without comorbid disorders, 16 with dyspraxia following preterm birth, 106 with motor difficulties (MD) without DCD, 51 comparison subjects with neurodevelopmental conditions other than DCD, and 1,213 typically developing (TD) controls, for a total of 2,766 subjects. MD groups were composed of children that had been identified with motor difficulties by scoring below the 16th percentile on the MABC-2 [47, 57–59] or had a standardized score under 80 on the McCarron Assessment of Neuromuscular Developmental [60], without having a diagnosis of DCD [61]. Some articles used the same tasks in the same sample of participants with DCD, and therefore, whether it was mentioned in the article [47, 58] or obvious (common authors, close in time, same sample size, characteristics and inclusion/exclusion criteria) [62–64], their samples were considered as one and their participants were counted only once in the reported total participants. The age range for all 41 studies combined is 3 to 17 years old and the mean is 9.46 years old. Results and characteristics of studies included in this review are presented in Table 1. The quality ratings ranged between 5 and 10 out of 10. Table 2 provides the details regarding quality assessment of all included studies. Seven studies (17.1%) needed to be discussed between two authors to reach a consensus about their quality level of evidence. It was finally rated as high in 18 studies, medium in 21 studies and low in two studies. The two studies that were rated as low quality were nevertheless included, because they still met the eligibility criteria established by authors. However, they are identified in the sections in which they are discussed in order to nuance their contribution.

No study explicitly assessing organization, self-regulation or goal-setting processes and meeting eligibility criteria established in this review was found. Overall, included studies explored nine dimensions of cognition: three attentional functions (alertness and sustained attention, selective attention, and divided attention) and six executive functions (inhibitory control, working memory, planning, cognitive flexibility, fluency and general executive functioning). A study was considered to discuss general executive functioning when the task they used intentionally assessed multiple executive components, by using an ecological task or a

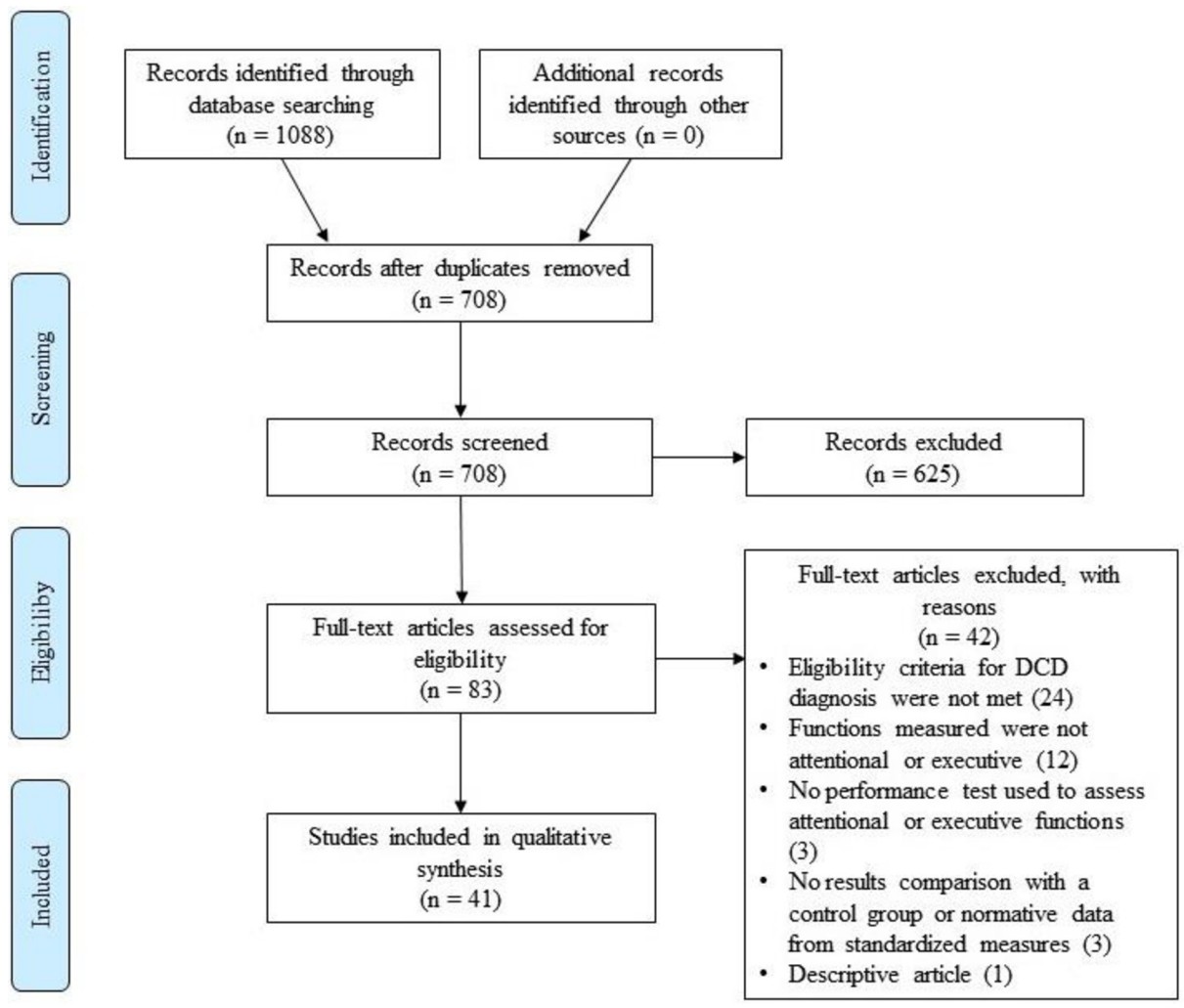

**Fig 1. PRISMA flow chart illustrating articles identification and selection process.**

standardized task that did not allow isolating one component of executive functioning. Results are presented in terms of these nine cognitive domains. When several components of attentional and executive functioning were assessed in the same study, the results for each component were reported in their respective section. When it was mentioned in the study, the different modalities (auditory/verbal or visual/nonverbal) used to measure each cognitive function are discussed separately (when available, the modality is identified in Table 1). When it was not possible to identify the modalities, the general cognitive function of interest is discussed. An exhaustive summary of attentional and executive functions assessed and their modalities, tasks used, sample size and age group in which they were respectively assessed, along with studies providing results about each cognitive domain, is provided in Appendix 2.

## 3.2. Attentional functions

**3.2.1. Alertness and sustained attention.** Among the seven studies that assessed alertness and sustained attention in the visual modality, four found no significant difference between performances of children with DCD and their TD peers or normative data [65–68], while

**Table 2. Quality assessment of studies included in the systematic review.**

| Studies Authors (year) | Quality items | | | | | | | | | | # of "yes" Quality level |
|---|---|---|---|---|---|---|---|---|---|---|---|
| | 1. Diagnosis and inclusion criteria mentioned | 2. Exclusion criteria mentioned | 3. Samples source mentioned | 4. Groups come from same community | 5. Groups are age matched | 6. Groups are gender matched | 7. Study controls for IQ | 8. Measures of outcome clearly described | 9. Adequacy of outcome analysis | 10. Implications and limitations discussed | |
| Alesi et al. (2019) | Yes | Yes | Yes | Yes | Yes | Yes | No | Yes | Yes | No | 8 High |
| Alloway (2007) | Yes | No | Yes | NA | NA | NA | Yes | Yes | Yes | No | 8 High |
| Alloway (2011) | Yes | Yes | Yes | Yes | Yes | No | Yes | Yes | Yes | No | 8 High |
| Alloway & Archibald (2008) | Yes | Yes | Yes | Yes | Yes | No | No | Yes | Yes | No | 7 Medium |
| Alloway et al. (2009) | Yes | Yes | No | No | No | No | Yes | Yes | Yes | Yes | 6 Medium |
| Alloway & Temple (2007) | Yes | No | Yes | Yes | Yes | No | Yes | Yes | Yes | No | 7 Medium |
| Asonitou & Koutsouki (2016) | Yes | Yes | Yes | Yes | Yes | Yes | No | Yes | Yes | Yes | 9 High |
| Asonitou et al. (2012) | Yes | Yes | Yes | Yes | Yes | No | No | Yes | Yes | Yes | 8 High |
| Barray et al. (2008) | Yes | Yes | Yes | Yes | Yes | No | Yes | Yes | Yes | No | 8 High |
| Bernardi et al. (2017) | Yes | Yes | Yes | No | No | No | Yes | Yes | Yes | Yes | 7 Medium |
| Bernardi et al. (2016) | Yes | Yes | Yes | No | No | No | Yes | Yes | Yes | No | 6 Medium |
| Biotteau et al. (2017) | Yes | Yes | Yes | Yes | Yes | No | No | Yes | Yes | Yes | 8 High |
| Blais et al. (2017) | Yes | Yes | Yes | No | Yes | No | No | Yes | Yes | No | 6 Medium |
| Chen et al. (2012) | Yes | Yes | Yes | No | Yes | No | No | Yes | Yes | No | 6 Medium |
| de Castelnau et al. (2007) | Yes | Yes | Yes | No | Yes | No | No | Yes | Yes | No | 6 Medium |
| Dyck & Piek (2010) | Yes | Yes | No | No | No | No | No | Yes | Yes | Yes | 5 Low |
| Gonzalez et al. (2016) | Yes | Yes | Yes | No | Yes | No | No | Yes | Yes | Yes | 7 Medium |
| Kaiser & Albaret (2016) | Yes | Yes | Yes | No | No | No | No | Yes | Yes | Yes | 6 Medium |
| Kirby et al. (2010) | Yes | Yes | Yes | Yes | Yes | No | No | No | Yes | Yes | 7 Medium |
| Leonard et al. (2015) | Yes | Yes | Yes | No | No | No | Yes | Yes | Yes | No | 6 Medium |
| Mandich et al. (2002) | Yes | Yes | Yes | No | Yes | No | No | Yes | Yes | No | 6 Medium |
| Mandich et al. (2003) | Yes | No | Yes | No | Yes | No | Yes | Yes | Yes | No | 6 Medium |

(*Continued*)

**Table 2.** (Continued)

| Studies Authors (year) | Quality items | | | | | | | | | | # of "yes" Quality level |
|---|---|---|---|---|---|---|---|---|---|---|---|
| | 1. | 2. | 3. | 4. | 5. | 6. | 7. | 8. | 9. | 10. | |
| | Diagnosis and inclusion criteria mentioned | Exclusion criteria mentioned | Samples source mentioned | Groups come from same community | Groups are age matched | Groups are gender matched | Study controls for IQ | Measures of outcome clearly described | Adequacy of outcome analysis | Implications and limitations discussed | |
| Piek et al. (2007) | Yes | Yes | Yes | No | No | No | Yes | Yes | Yes | No | 6 Medium |
| Pratt et al. (2014) | Yes | Yes | Yes | No | Yes | No | Yes | Yes | Yes | No | 7 Medium |
| Querne et al. (2008) | Yes | Yes | No | No | Yes | No | Yes | Yes | Yes | Yes | 7 Medium |
| Rahimi-Golkhan-dan et al. (2016) | Yes | Yes | Yes | Yes | Yes | No | No | Yes | Yes | Yes | 8 High |
| Ruddock et al. (2016) | Yes | Yes | Yes | Yes | Yes | No | No | No | Yes | Yes | 7 Medium |
| Ruddock et al. (2015) | Yes | Yes | Yes | Yes | Yes | No | No | No | Yes | Yes | 7 Medium |
| Sartori et al. (2020) | Yes | No | Yes | Yes | Yes | Yes | No | Yes | Yes | No | 7 Medium |
| Sumner et al. (2016) | Yes | Yes | Yes | No | Yes | Yes | No | Yes | Yes | Yes | 8 High |
| Thrornton et al. (2018) | Yes | Yes | Yes | Yes | Yes | No | Yes | Yes | Yes | No | 8 High |
| Toussaint-Thorin et al. (2013) | Yes | Yes | No | No | Yes | No | No | Yes | No | Yes | 5 Low |
| Tsai et al. (2012) | Yes | Yes | Yes | Yes | Yes | Yes | Yes | Yes | Yes | Yes | 10 High |
| Tsai et al. (2010) | Yes | Yes | Yes | Yes | Yes | Yes | Yes | Yes | Yes | No | 9 High |
| [1]Tsai et al. (2009) | Yes | Yes | Yes | Yes | Yes | No | Yes | Yes | Yes | No | 8 High |
| [2]Tsai et al. (2009) | Yes | Yes | Yes | Yes | Yes | Yes | Yes | Yes | Yes | No | 9 High |
| Wang et al. (2015) | Yes | Yes | Yes | Yes | Yes | Yes | Yes | Yes | Yes | No | 9 High |
| Williams et al. (2013) | Yes | Yes | Yes | No | No | No | Yes | Yes | Yes | Yes | 7 Medium |
| Wilson & Maruff (1999) | Yes | Yes | Yes | Yes | Yes | Yes | Yes | Yes | Yes | No | 9 High |
| Wilson et al. (1997) | Yes | Yes | Yes | Yes | Yes | Yes | Yes | Yes | Yes | No | 9 High |
| Zhu et al. (2012) | Yes | Yes | Yes | No | Yes | Yes | No | Yes | Yes | Yes | 8 High |

Note: NA = Non-applicable; was counted as "yes" to avoid penalizing studies for criteria that did not apply to them.

three studies reported that children with DCD had significantly more difficulty than TD children in maintaining their alertness during a long, boring visual task [44, 69, 70]. In the auditory modality, children with DCD did not differ from their TD peers [71].

Regarding the presence of comorbid disorders, (for more details about exclusions and comorbid disorders present in the samples, see Table 1), the three studies that compared the performance of children in different clinical groups on tasks of alertness or sustained attention found that children with DCD, DDL or both DCD and DDL did not differ in their sustained attention capacities [65], nor did children with a single diagnosis of either DCD or ADHD [66]. However, the study by Williams et al. [71] revealed that children with a single diagnosis of DCD do not exhibit a deficit in auditory sustained attention compared to TD controls, but that participants with comorbid DCD and ADHD did have greater difficulty than the normative group in this domain. This suggests that overall, DCD in itself does not seem to be associated with deficits in alertness or sustained attention, but rather the co-occurrence of ADHD, which is common in this population, negatively influences these abilities.

**3.2.2. Selective attention.** Among the four articles that discussed selective attention in the visual modality, two reported impairment on this attentional domain in children with DCD [72, 73]. The two other studies did not find any significant difference between performance of children with DCD or DD and TD controls or normative data [66, 74]. However, it seems that children with dyspraxia following preterm birth had significantly poorer scores on a visual attention task than children with DD, a difference that may be due to sequalae related to prematurity rather than dyspraxia [74]. In addition, both studies that assessed auditive selective attention in children with DCD reported no deficit [46, 74]. Even though the results of both studies are consistent, note that the study by Toussaint-Thorin et al. [46] was the one qualified as low quality and, for this reason, their results should be considered cautiously.

Regarding the management of possible comorbid neurodevelopmental disorders in children composing the DCD groups, the two studies that excluded children with any other medical, neurological or developmental conditions [72, 73] were the only studies to find deficits in selective attention in children with DCD. Thus, these impairments do not appear to be linked to the presence of comorbid conditions. The reasons why studies including children with co-occurring disorders in their DCD sample did not find deficits in this domain may include methodological weaknesses, since the study by Toussaint-Thorin et al. [46] was rated as low quality, but further research is needed to elucidate this point. Finally, Kaiser and Albaret [66] compared children with DCD and children with ADHD on this attentional domain, and they found no significant difference between their performances, suggesting that these clinical groups cannot be distinguished based on their selective attention capacities.

**3.2.3. Divided attention.** Results showed no impairment on this attentional component in participants with DCD when compared to TD controls, and there was no significant difference between DCD and ADHD groups [66]. The authors only mention excluding children with ADHD from their DCD sample and thus, it is possible that at least some children presented other comorbid neurodevelopmental conditions or learning difficulties. However, if present, it does not seem that these concomitant disorders had a negative influence on the children's divided attention capacities, since no deficits were found.

## 3.3. Executive functions

**3.3.1. Inhibitory control.** Two components of inhibitory control were assessed: the response inhibition component and the attentional inhibition component.

*3.3.1.1. Response inhibition.* Overall, 17 studies measured nonverbal response inhibition, while verbal response inhibition was assessed in five different samples.

**3.3.1.1.1. Nonverbal response inhibition.** Difficulties in children with DCD regarding the ability to inhibit a nonverbal prepotent response when compared to their TD peers were found in ten different studies, especially in terms of correct responses [46, 47, 57–59, 67, 69, 75–78].

Rahimi-Golkhandan et al. [67] specified that the response inhibition deficit in DCD children is only present when stimuli are positively-valenced, and thus more compelling. However, another study using emotionally-valenced tasks found no significant difference between children with DCD and TD controls [79]. Finally, five studies using independent samples found no deficit on this ability in participants with DCD or MD compared to normative data or a control group [44, 61, 65, 70, 80].

Regarding comorbid disorders, Thornton et al. [81] found that only children with co-occurring DCD and ADHD made significantly more errors than their TD peers, while there was no significant difference between performance of children with a single diagnosis of DCD or ADHD and TD controls.

*3.3.1.1.2. Verbal response inhibition.* In one of the five samples, the difference between DCD and TD groups was significant on the reduced motor-load task (verbal task), while they were not on the high motor-load task (nonverbal task) [80]. However, the authors propose that these results might be better explained by the tasks' complexity rather than by their modality. Two other samples found a significant impairment as children with DCD made more errors than TD children on tasks requiring a verbal response [59, 79]. In the two other samples, no impairment was found on verbal inhibition measures in terms of accuracy [57, 58], but Bernardi et al. [58] found that participants with DCD were significantly slower than their TD peers on the verbal tasks. Longer completion time on verbal response inhibition tasks has also been found in two other studies [59, 79].

To summarize, children with DCD exhibit response inhibition difficulties in the nonverbal modality, as well as in the verbal modality of response inhibition.

Furthermore, response inhibition difficulties were found in children with DCD, whether they had comorbid conditions or not. In addition, Biotteau et al. [65] compared the results of children with DCD with those of children with DDL and found no significant difference between children with DCD only, DCD with comorbid DDL, or DDL only on these capacities. Hence, impairments that were found in most of the studies included in this subsection do not appear to be due to the presence of comorbid neurodevelopmental disorders.

*3.3.1.2. Attentional inhibition.* Two different modes of orienting attention were studied: the endogenous and the exogenous mode. The former is defined as a controlled and volitional allocation of attentional resources requiring a cognitive interpretation of stimuli, while the latter refers to an automatic and reflexive allocation of attention that has an alerting utility [82, 83].

All six articles that assessed the endogenous mode of orienting attention reported a deficit of endogenous attentional inhibitory control or "disengagement inhibition" of attention in children with DCD, when compared to TD controls [75, 82–86]. Among the seven studies discussing the exogenous mode, four found no deficit [76, 82, 83, 85], while three reported this mode to be impaired in children with DCD [87–89]. The lack of consistency between studies could be related to the nature of the precues used in certain tasks (eyes) compared to the ones previously used (arrows), which presumably trigger different brain areas and neural networks, may be more alerting and involve volitional components of orienting attention as well as automatic components [88, 89]. In summary, the endogenous aspects of attentional inhibition seem to be more severely affected in children with DCD than exogenous aspects, which means these children have more difficulty to voluntarily direct and shift their attention toward a stimulus or a task while inhibiting distractors than to automatically shift their attention to peripheral alerting stimuli.

Regarding the presence of comorbid disorders in the DCD samples, deficits in attentional inhibition were found in children with pure DCD as well as in subjects with possible co-occurring disorders, suggesting that impairments in this domain seem to be an inherent part of DCD and do not seem to be attributable to other disorders associated with DCD.

**3.3.2. Working memory.**    The visuospatial modality of working memory and its verbal modality were assessed.

3.3.2.1. Visuospatial working memory. All eight samples of children with DCD showed impairments of visuospatial working memory compared to normative data or control groups [26, 47, 57, 59, 63, 68, 79, 90]. Children with MD without DCD did not differ from those with DCD: both groups performed more poorly than TD children [47, 59].

As for the comorbid neurodevelopmental disorders, Alloway [63] and Alloway et al. [64] found no significant difference between children with DCD and children with ADHD, SLI or ASD. In fact, children with ASD performed better than those with DCD, but the difference was not significant once the nonverbal IQ test's shared motor component was accounted for [64]. Furthermore, a study that compared two groups of children with DCD, one in which children had language or nonverbal reasoning difficulties and one in which their disorder was purer, found no difference on visuospatial working memory capacities between the two groups. However, children with a purer DCD performed significantly worse than children with SLI in this study, even when the contribution of receptive language skills was accounted for [26]. Also, children with DCD were significantly more impaired than children with general learning difficulties on the visuospatial working memory modality [90]. Considering these results, visuospatial working memory seems to be a deficit in children with DCD whether their disorder is pure or not, and it appears to be more specific to children with DCD than to children with mild learning impairments or SLI. However, this deficit is similar in children with DCD, ADHD or ASD.

*3.2.2.2.* Verbal working memory. Three samples showed no impairment of verbal working memory in children with DCD when compared to TD controls or normative data [47, 57, 65]. Children with MD without DCD did not differ from children with DCD nor TD children [47, 59]. However, eight different samples showed difficulties in verbal working memory in children with DCD according to comparisons with control groups or normative data [20, 26, 59, 61, 63, 79, 90, 91]. Again, children with poor motor coordination without DCD did not differ significantly from children with DCD [61]. However, the study by Dyck and Piek [61] was rated as low quality because of the non-comparability of their groups, and thus these results might be explained by confounding variables. Moreover, two research teams put their results in perspective: Sumner et al. [91] mentioned they could not conclude to a primary deficit in verbal working memory in children with DCD since their results showed a great heterogeneity across all intelligence domains in these children, while Piek et al. [20] explained that their results could be attributable to a slower processing speed in children with DCD, although further analysis would be required to confirm this hypothesis.

Regarding the presence of comorbid disorders, among the eight different samples in which verbal working memory difficulties were found, all comorbid conditions were excluded from four samples [59, 61, 79, 91]. Consequently, it seems possible that the impairments found in some studies are, at least in part, influenced by comorbid disorders. Furthermore, among the 11 articles included in this section, nine compared the results of children with DCD on verbal working memory measures with those of children with non-motor neurodevelopmental or learning problems. While the five studies finding at least some difficulties in children with DCD on verbal working memory measures reported no difference between children with DCD and children with ADHD, SLI, ASD, mild learning difficulties, mixed receptive expressive language disorder (RELD) or relatively poor language ability [26, 61, 63, 64, 90], only one found that children with DCD performed significantly poorer than children with ADHD [20]. Among studies that did not report any impairment in verbal working memory, one specified that there was no difference between their two groups of DCD subjects, one in which children had a diagnosis of comorbid DDL and one purer group, nor with a group of children with a

single diagnosis of DDL [65]. In summary, impairment of verbal working memory in children with DCD is less clear than the deficit observed in visuospatial working memory, which, however, does not seem to be specific to DCD.

**3.3.3 Planning.** Two studies compared children's performance on nonverbal and verbal measures [47, 65], while six considered general planning abilities [46, 72–74, 80, 92]. Among the six articles that discussed planning irrespective of the task's modality, only one reported no deficit on this ability compared to normative data [74], while the other five reported significantly poorer performance in children with DCD in comparison to TD children [46, 72, 73, 80, 92]. One of these five studies was however rated as low quality [46]. Pratt et al. [80] specify that the task's motor-load did not affect performance on the planning measures, since subjects with DCD were impaired in both tasks, even when the effect of perceptual reasoning was accounted for. Also, the two studies that compared performance on nonverbal and verbal measures reported diverging results: Leonard et al. [47] found no difference between children with DCD and TD controls on both nonverbal and verbal measures, while Bernardi et al. [57] reported that children with DCD, as well as children with MD, performed more poorly than TD controls on the nonverbal measure of planning, but observed no significant difference on the verbal measure. Thus, in all children with motor problems, planning might be impaired on a nonverbal measure, but not on a verbal one.

Regarding the presence of comorbid disorders in DCD samples of studies included in this section, among the six articles that reported at least some impairment of planning in children with DCD four used pure samples [57, 72, 73, 80]. Therefore, impairments found do not seem to be attributable to the presence of comorbid disorders, and these do not appear to influence planning abilities in children with DCD.

**3.3.4. Cognitive flexibility.** Among the three articles that addressed general cognitive flexibility, one found no deficit [46], while the other two reported significantly worse performance in children with DCD than in controls [20, 59]. Furthermore, Sartori et al. [59] found no difference between DCD children and children with MD without DCD. Note, however, that the study reporting no deficit was rated as low quality, and thus more weight should be given to the studies by Piek et al. [20] and Sartori et al. [59]. In regard to the two studies that compared performance on nonverbal and verbal measures, Leonard et al. [47] found no difference between groups of participants with DCD, MD without DCD, and TD controls on both nonverbal and verbal measures, while Bernardi et al. [57] reported that children with DCD performed more poorly than TD controls on the nonverbal measure of cognitive flexibility, without any difference between the MD and the TD groups. Cognitive flexibility was thus impaired specifically in children with DCD, but only on the nonverbal measure. However, it is important to note that both studies used a Trail Making test as a measure of cognitive flexibility, which is a graphomotor task. It is thus possible that the impairment reported by Bernardi et al. [57] is at least partly attributable to the primary motor deficits of children with DCD.

Regarding the presence of comorbid disorders among DCD samples, two out of three studies that used a pure sample found impairments in cognitive flexibility [57, 59], specifying that difficulties are only present on a nonverbal measure. One study that included children with comorbid conditions did not find any difficulties [46], while the other found some [20]. Furthermore, one study compared the performance of children with DCD with that of children with ADHD and found that the firsts were more impaired than the seconds on a general cognitive flexibility measure [20]. Since results are heterogeneous, no specific cognitive flexibility profile seems to be associated with DCD, although weaker flexibility in the nonverbal domain might be present. In addition, comorbid disorders do not seem to have a notable influence on this ability in this clinical group.

**3.3.5. Fluency.** The nonverbal and verbal modalities of fluency were assessed.

*3.3.5.1. Nonverbal fluency.* All three studies reported difficulties in children with DCD compared to normative data or control groups [47, 57, 74]. Children with MD without DCD were at par with children with DCD, both groups showing poorer performance than TD children [47]. Furthermore, there seems to be no difference between children with DCD and children with dyspraxia following preterm birth on nonverbal fluency, but Barray et al. [74] pointed out that the difficulties observed in these clinical groups on design fluency tasks could be attributable to graphomotor difficulties rather than fluency weaknesses per se [74].

Regarding the presence of comorbid disorders among the DCD samples, whether neurodevelopmental comorbidities are excluded [47, 57] or not [74], nonverbal fluency capacities appear to be impaired in children with DCD.

*3.3.5.2 Verbal fluency.* Two studies found no deficit on verbal fluency in children with DCD or DD compared to control group [46, 47], while the other two found that children with DCD performed more poorly than controls [57, 79]. However, since the study by Toussaint-Thorin et al. [46] was rated as low quality, their results should be considered with caution.

Regarding the presence of comorbid disorders among the children with DCD or DD, exclusion [47, 57, 79] or inclusion [46] of ADHD did not seem to influence the results about verbal fluency capacities.

**3.3.6. General executive functioning.** Both studies reported that children with DCD had significantly lower performance than a control group composed of healthy individuals [46, 93]. Children with DCD were impaired on most measures of the WCST [93], and they made more errors, exhibited difficulties in respecting the task's guidelines and were more dependent on the cooking task, suggesting impairments in problem solving abilities [46]. However, since this study is considered of low-quality, more research using ecological tasks is needed to confirm its conclusions.

Regarding the presence of comorbid neurodevelopmental disorders, Zhu et al. [93] excluded participants with any neurodevelopmental or medical conditions, while Toussaint-Thorin et al. [46] included children with ADHD in their DCD sample. Thus, children with DCD seem to exhibit difficulties in tasks integrating multiple components of executive functioning, whether they also have ADHD or not.

## 3.4. Developmental considerations

The effect of age on cognitive functions has been studied in visual sustained attention, response inhibition, working memory, planning, cognitive flexibility and fluency abilities.

Regarding visual sustained attention, in a cross-sectional study, de Castelnau et al. [70] found that this capacity improved with age in both their sample of children with DCD and their TD group, but the discrepancy between the two groups remained, from 8–9 years old to 12–13 years old. These results suggest that difficulties in sustained attention persist with age in children with DCD.

Four studies considered the effect of years passing on response inhibition capacities. Three studies using a cross-sectional design found that nonverbal response inhibition capacities were better in older participants with DCD than in younger children with the disorder [70, 77, 78], so that these capacities in children with DCD approached those of TD children as they got older [77, 78]. The other study, using a longitudinal design, found no improvement in nonverbal response inhibition in a DCD sample after a two-year follow-up, and the differences originally found between their DCD group and their TD group persisted two years later [57]. These results indicate that the impairment found in nonverbal response inhibition in children might be due to a delay in the development of this cognitive function rather than a primary deficit, but given the heterogeneity of results across studies, more research is necessary to confirm this

hypothesis. Regarding verbal inhibition capacities, Bernardi et al. [57] reported no significant change after two years in their DCD sample, although there was no deficit in DCD at both time points.

Additionally, among the functions that they found to be impaired in children with DCD in their longitudinal study, Bernardi et al. [57] reported a significant improvement of visuospatial working memory, nonverbal planning, nonverbal cognitive flexibility and nonverbal and verbal fluency over time in children with DCD. They also mention that improvements in their TD group were similar, so that the deficit found in these domains in children with DCD, in comparison to TD children, persisted after a two-year follow-up [57].

In summary, attentional and executive functions seem to improve with age in children with DCD, as well in TD controls, but studies found that discrepancies between performance of children with DCD and that of TD children persist over time on most executive tasks.

## 4. Discussion

### 4.1. Summary of findings

The 41 articles included in this systematic review, totalizing 1,097 children and adolescents with DCD, repeatedly reported significant impairment of nonverbal response inhibition, attentional inhibition, visuospatial and verbal working memory, planning, nonverbal fluency and general executive functioning in children with DCD. Studies assessing cognitive flexibility are divided on the presence or magnitude of impairments: one study, with a relatively small sample size (n = 17 DCD), found a deficit only on a nonverbal measure, while another study, with 23 subjects with DCD, did not find any significant impairment. Similarly, half of the studies measuring verbal response inhibition found some impairment in this area, although completion time was the only aspect affected in one study. Regarding studies exploring alertness and sustained attention, selective attention, divided attention and verbal fluency, most found no deficit in these areas in children with DCD. No studies investigating organization and self-regulation skills were included in this systematic review as none met the inclusion criteria.

In light of these results, there is no evidence of an obvious attentional deficit in children with DCD, but executive functions are widely impaired. Namely, difficulties are not limited to domains directly associated with core impairments of the disorder (e.g. nonverbal planning), but extend to other areas of executive processes, including nonverbal inhibitory control, nonverbal and verbal working memory, and general planning abilities, as well as on ecological tasks integrating multiple components of executive functions. In summary, executive functions are more impaired on a nonverbal/visuospatial modality than on a verbal modality, and results suggest that a relatively broad executive deficit is present in children with DCD.

A few studies also explored the possible differences between a group of children with DCD and a group of children with MD without DCD. As previously described, these groups were composed of children whose motor difficulties have been objectified, but who did not meet formal diagnostic criteria for DCD. It is interesting to note that in most studies, no significant difference was noted between MD and DCD groups in terms of executive impairments. This raises the possibility that executive difficulties are tightly linked with motor impairments, irrespective of severity. Consequently, the findings and conclusions about DCD highlighted in this review could also apply to a large portion of children with measurable motor difficulties that do not have a clinical diagnosis of DCD.

### 4.2. Influence of comorbid disorders

Among the cognitive functions that were noted as being impaired in most studies, the presence of comorbid disorders did not seem to have a determinant influence on inhibitory control

(both response and attentional inhibition), visuospatial working memory, planning, cognitive flexibility, nonverbal fluency and general executive functioning capacities. Thus, the weaknesses described in these domains do not appear to be attributable to co-occurring disorders, but rather suggest that cognitive impairments are integrant elements of DCD. This is unexpected since DCD is primarily a motor disorder, and the definition does not include executive functioning problems [1–3]. However, not all studies reviewed here addressed the issue of comorbidities in the same way: some studies based their exclusion criteria only on comorbid disorders that had been diagnosed in children, while others measured and confirmed their presence in children composing their sample. In most studies in which response inhibition, visuospatial working memory, planning, cognitive flexibility, nonverbal fluency and general executive functioning were assessed, children with co-occurring disorders were excluded based on the presence of previously diagnosed disorders, without these diagnoses being confirmed by the research teams. However, the presence of children's difficulties linked to comorbid disorders was confirmed by authors in most studies investigating attentional inhibition. Therefore, it is particularly clear that deficits in attentional inhibition are part of DCD. However, since most studies did not confirm or refute the presence of co-occurring disorders, it is possible that some children in DCD samples nonetheless suffered from additional neurodevelopmental disorders that were not documented. Considering this, additional studies using large sample size and thorough screening methods to assess the presence of comorbid disorders are needed to conclude on the definite influence of comorbid conditions on the cognitive profile of children with DCD.

Despite the imprecision of screening and sometimes inclusions and exclusion criteria in some studies, it seems likely that comorbid conditions exert at least some influence on verbal working memory capacities. Indeed, among the eight samples in which verbal working memory difficulties were found, all comorbid disorders were excluded from four samples [59, 61, 79, 91], and two other samples were free of ADHD, behavioral problems or ASD [26, 63]. It is thus unlikely that the impairments reported in these studies are due to these co-occurring neurodevelopmental disorders, but it does not exclude the possible influence of language or learning difficulties in half of the studies [20, 26, 63, 90]. Interestingly, among the three samples in which verbal working memory abilities were preserved, two were pure DCD samples and one was free of comorbid ADHD or SLI. In summary, when at least the co-occurrence of SLI is excluded, impairments in verbal working memory are less probable in children with DCD. Therefore, deficits in this area might not be part of the DCD profile per se, but may appear when co-occurring difficulties, especially in terms of language, are present.

It appears that difficulties observed on measures of visuospatial and verbal working memory are frequent but not specific to children with DCD, since they are not significantly different than those found in children with neurodevelopmental disorders that do not necessarily involve a motoric component, including ADHD, SLI, RELD, ASD, and learning difficulties. For this reason, it is possible that impairments in these domains could reflect a general sign of an atypical cognitive development, without being specific to a given disorder, nor systematically present in children with neurodevelopmental conditions.

## 4.3. Brain correlates

Studies on the neural correlates of DCD contribute to our understanding of the results included in the present review. Several neuroimaging studies have revealed implications of corpus callosum, basal ganglia, inferior parietal cortex, thalamus and cerebellum in DCD, and of connections between these structures [44, 65, 94, 95]. Abnormalities have also been noted in frontal cortex functioning and in white matter maturation and composition in individuals

with DCD [44, 95–97]. Furthermore, deficits in functional activation during a motor response inhibition task have been found in the right pre- and postcentral gyri, as well as in the left superior and middle frontal gyri [81]. However, the authors specified that these deficits were only present in children with co-occurring DCD and ADHD. Given the plurality of brain areas involved in DCD, the hypothesis of an "Atypical Brain Development" (ABD) has been proposed. This view stipulates that brain abnormalities underlying the cognitive and motor deficits observed in various neurodevelopmental disorders are rather diffuse than localized [19, 98]. ABD is not a disorder per se, but rather a suboptimal developmental process that manifests itself in different forms, such as motor, attentional and/or reading disorders [19, 99]. Given the strong overlap between symptoms of DCD and ADHD [19], the umbrella term "Deficit in Attention, Motor control and Perception" (DAMP) was formerly proposed to define a combination of DCD and ADHD [6, 100]. The authors insisted on the importance of a term acknowledging both attentional and motor control problems in clinical practice, stipulating that these symptoms have strong common background factors and that the prognosis of children with DAMP was poorer than that of children with either DCD or ADHD [100].

Several authors have also hypothesized that children with DCD could have an automatization deficit, related to a cerebellar dysfunction [19]. Although possibly greater and more obvious in children with DCD than in children with ADHD and/or DDL, this deficit does not appear to be specific to the DCD population, but is on the contrary common to most developmental disorders [19, 101]. It could therefore explain the high prevalence of comorbidities among neurodevelopmental disorders and the non-specificity of the cognitive impairments found amongst them [98, 102].

The fact that the frontal cortex, basal ganglia, cerebellum and parietal cortex are especially implicated in attentional and executive functioning [4, 38, 41], and that these brain structures do not operate optimally in individuals with DCD, as previously described, might explain some of the attentional and executive impairments compiled in the present review. Additionally, an atypical hemispheric lateralization for attention and inhibitory functions has been shown in children with DCD, along with a reduced efficiency of cerebral network involved in inhibitory control [44]. Evoked potential studies have also reported that children with DCD allow fewer resources than TD controls for spatial locations comparison and response retrieval and selection [68], and that they may have less mature anticipatory and executive processes, reduced interhemispheric and cognitive-to-motor transfer speeds as well as an atypical neural activity associated with attentional control [85, 88, 89]. Without providing correlational or causal evidence, the set of cerebral abnormalities observed in DCD, which involves key structures robustly involved in the corticostriatal function and the frontal cortex, the presence of impairments in the executive and attentional domain in DCD is not unexpected.

## 5. Conclusion

This systematic review of the literature revealed that children and adolescents with DCD show impairments mostly on tasks of inhibitory control, working memory, planning, nonverbal fluency and general executive functioning. Alertness and sustained attention, selective and divided attention and verbal fluency capacities appear more intact, whereas results regarding cognitive flexibility are divided. More evidence supports the presence of a deficit in nonverbal executive tasks, without the impairments being exclusive to this modality. Co-occurring disorders might influence impairments found on verbal working memory capacities in children with DCD.

These results contribute to a better understanding of the cognitive profile associated with DCD and have several implications. On a conceptual level, we must consider that executive

impairments are common in DCD and, although they may be partly explainable by the underlying visuospatial and motor deficits, they are not entirely so, especially in areas of working memory, planning abilities and general executive functioning. When evaluating executive functioning in children with DCD, clinicians should still be aware of the possible contribution of visuospatial and motor deficits in their results, and impairments on executive tasks that do not require visuospatial or motor skills should be found before concluding to an executive deficit in a child with DCD. Since a few studies have identified similar difficulties in children with DCD to children with less severe motor difficulties that did not have a diagnosis of DCD, clinicians must keep in mind that children with motor difficulties that have not been diagnosed with DCD may present the same set of executive difficulties. Furthermore, when clinicians do conclude to executive deficits in a child with DCD, they must keep in mind that attentional and executive impairments may be part of the disorder itself. While it is important that children benefit from appropriate interventions to help them with their difficulties, professionals should also be parsimonious in diagnosing concomitant disorders. Indeed, it appears possible that the high prevalence of comorbid disorders found in children with DCD is due to cognitive dysfunction related to DCD itself, without necessarily being specific to this disorder, rather than to an additional disorder. In all cases, professionals intervening with children and adolescents with DCD should expect them to be impulsive and to be easily distracted by task-irrelevant stimuli. They should also adapt their interventions to try to avoid overloading their working memory and support them in developing their planning skills.

## 5.1. Strengths and limitations of the current review

This systematic review of the literature has several strengths. On a methodological level, the facts that we used a broad variety of research terms for executive functions and that we included studies using different types of performance tasks, from experimental tasks to standardized neuropsychological tests, allow our review to be exhaustive, inclusive, and thus more representative of the attentional and executive functions in children and adolescents with DCD. The inclusion of studies using more ecological tasks is also a strength since our results contribute to a better understanding and representation of these children's daily difficulties. In addition, this review is the first to explicitly discuss the influence of concomitant disorders on attentional and executive processes in DCD. The fact that we included studies that used DCD samples with comorbid disorders, as well as the fact that the included studies recruited their DCD sample from various sources, increases the likelihood that the conclusions drawn are ecologically valid and generalizable to the population of children with DCD.

Regarding methodological limitations, the lack of search and inclusion of grey literature and nonpublished studies may bring a publication bias. Furthermore, a standardized quality assessment tool for neuropsychological studies could not be found, so we developed our own quality assessment checklist based on the NOS and the PRISMA standards. By this mean, this review raised the risk of including studies that provide inadequate quality of evidence. Although we explicitly mentioned in the relevant sections when the quality of a study was low, a common and important limitation of several studies was the small sample size, which entails that even when grouped together, the number of participants remained relatively limited. This is especially relevant for results regarding auditive alertness/sustained attention, divided attention, comparisons between verbal and nonverbal planning and between verbal and nonverbal cognitive flexibility, and verbal fluency. Similarly, some of the attentional and executive components discussed in this review have only been the topic of a small number of studies, particularly selective attention, divided attention, cognitive flexibility, fluency and general executive functioning. Thereby, we must be cautious when considering the conclusions drawn regarding

these domains, and further studies are necessary to confirm the interpretations made of these findings. Another important limitation of studies included in this review is the inconsistency regarding the clarity of exclusion criteria of several DCD samples. As the presence of comorbid disorders was not always rigorously documented nor confirmed, it is possible that comorbid disorders may play a larger role in the results highlighted in this review than we believe.

Other factors are to be considered when interpreting the conclusions of this systematic review. Firstly, we presented our results according to nine attentional and executive components to reduce the heterogeneity of the construct evaluated in each section. However, on a conceptual level, attentional and executive functions, although dissociable, are also united and interrelated [33, 36, 41, 103]. Similarly, all tasks require the implication of more than a single cognitive function, and in certain cases, it might be difficult to isolate one attentional or executive component. As a matter of fact, some studies using the same assessment tools reported their results in relation to different cognitive domains. Hence, the results described might be partially due to impairment in other cognitive areas than those purportedly assessed. Also, several tasks used to measure cognitive functions require a motor component. Children with DCD could perform more poorly on these tasks not because of an attentional or executive impairment, but because of their primary motor deficit associated with their disorder. Secondly, no study explicitly assessing organization, self-regulation or goal-setting processes were included since none met the eligibility criteria established in this review and because few tasks assessing these components of executive functions exist. Therefore, this systematic review does not allow discussing the integrity of these executive functioning components. Thirdly, even though eligibility criteria allowed including studies with subjects up to 17 years old, the most often studied ages were 8 to 12 years old. Given the considerable development of executive functions through childhood and adolescence [36, 38, 104], it would be incorrect to generalize the results of this review to younger children and older adolescents. Fourthly, it was previously mentioned that impairments found in the domains of visuospatial and verbal working memory were not specific to children with DCD, since children with ADHD or ASD shared these difficulties. Yet children with ADHD and ASD also often share sensorimotor deficits with children with DCD [23, 24, 105]. Given the overlap between symptoms that seems to be present, there is a limitation to the comparisons that can be made between these clinical groups. Finally, as in most neurodevelopmental disorders, there is great heterogeneity among children with DCD [19, 26, 65, 91]. The lack of details regarding the severity and predominant type of impairments could contribute to the variability of results noted in certain cognitive areas.

## 5.2. Directions for future research

Additional studies using larger samples and replicating the results of these studies are needed, especially in domains of alertness/sustained attention, divided attention, verbal and nonverbal planning, verbal and nonverbal cognitive flexibility, and verbal fluency. Likewise, a small number of studies have explored selective attention, divided attention, cognitive flexibility, fluency and general executive functioning in children in DCD, and more research on these cognitive functions is also needed to confirm the results highlighted in this review. Since no studies discussing organization, self-regulation and goal-setting skills and meeting our eligibility criteria were found, more rigorous studies exploring these executive functions are needed. Future studies should also focus on children under 8 years old and over 12 years old. Considering the development of attentional and executive functions during childhood and adolescence, and the variability of developmental trajectories among the different components of executive functioning [104], cross-sectional and longitudinal studies comparing different age groups and following children through several years would be important in order to determine

whether children with DCD present a lasting executive deficit or whether their difficulties are due to a developmental delay that catches up with time. In addition, to improve results generalization and comparison between studies, future research should avoid selecting their sample only in specialized clinical settings in order to obtain samples that are more representative of the population of children with DCD. Furthermore, given the fact that co-occurring disorders may have an influence on some executive functions, especially on verbal working memory, future studies should document and verify the presence of comorbid conditions in their sample. More studies should also compare attentional and executive functioning profiles of children with DCD without any other developmental or neurological problem, with that of children with other neurodevelopmental disorders and of children with multiple diagnoses to explore the possible distinction between these groups. Lastly, future research should allow better understanding of the overlap between executive functions and motor skills. To do so, the choice of tasks should be judicious and make possible a better comparison of children's performance on tasks requiring motor and visuospatial skills with their performance on tasks requiring no such abilities. In addition, given the overlap between DCD, ADHD and ASD previously discussed, a dimensional approach could be useful to go beyond group comparisons and allow a better understanding of co-occurring motor and cognitive difficulties. With more studies using larger samples, a meta-analytic approach would be relevant and could control for such confounding variables.

## Supporting information

**S1 Data. Minimal data set.**
(DOCX)

**S1 Checklist. PRISMA 2009 checklist.**
(DOC)

**S1 Appendix. Quality assessment: Adaptation of the Newcastle-Ottawa quality assessment scale cohort studies.**
(DOCX)

**S2 Appendix. Summary of attentional and executive functions assessed, age ranges, sample sizes and tasks used among studies.**
(DOCX)

## Author Contributions

**Investigation:** Mélodie Proteau-Lemieux.

**Methodology:** Catherine Lachambre, Mélodie Proteau-Lemieux, Sarah Lippé.

**Supervision:** Jean-François Lepage, Eve-Line Bussières, Sarah Lippé.

**Validation:** Catherine Lachambre.

**Writing – original draft:** Catherine Lachambre, Mélodie Proteau-Lemieux.

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
