## [Decision Letter · Decision Letter 0]

10 Feb 2021

PONE-D-20-33814

Attentional and Executive Functions in Children and Adolescents with Developmental Coordination Disorder and the Influence of Comorbid Disorders: A Systematic Review of the Literature

PLOS ONE

Dear Dr. Lippé,

Thank you for submitting your manuscript to PLOS ONE. After careful consideration, we feel that it has merit but does not fully meet PLOS ONE’s publication criteria as it currently stands. Therefore, we invite you to submit a revised version of the manuscript that addresses the points raised during the review process.

In view of the criticism of the reviewers found at the bottom of the page, your manuscript has been denied publication.  I too have read your manuscript and agree with the reviewers’ observations that is necessary to improve:

The research question should be added clearly in the introduction. Also, the objective that appears in the summary does not match (or is misleading) with what appears as a research question in the objectives. It would be interesting to improve this aspect.

It is recommended to relate the content of the discussion with what was previously stated in the introduction and avoid explaining future prospects in this section.

In this sense, it is desirable that the conclusions section appears before Strenghts and Limitations and Directions for Future Research.

We look forward to receiving your revised manuscript.

Kind regards,

Celestino Rodríguez, PhD

Academic Editor

PLOS ONE

Journal Requirements:

2) Thank you for stating the following in the Acknowledgments Section of your manuscript:

[JFL is supported by a Junior I Research Scholar Award from the Fonds de la Recherche

du Québec – Santé. MLP is supported by a scholarship from the Faculté de Médecine

et des Sciences de la Santé of Université de Sherbrooke]

 [The author(s) received no specific funding for this work]

3) PLOS requires an ORCID iD for the corresponding author in Editorial Manager on papers submitted after December 6th, 2016. Please ensure that you have an ORCID iD and that it is validated in Editorial Manager. To do this, go to ‘Update my Information’ (in the upper left-hand corner of the main menu), and click on the Fetch/Validate link next to the ORCID field. This will take you to the ORCID site and allow you to create a new iD or authenticate a pre-existing iD in Editorial Manager. Please see the following video for instructions on linking an ORCID iD to your Editorial Manager account: https://www.youtube.com/watch?v=_xcclfuvtxQ

4) Please include captions for your Supporting Information files at the end of your manuscript, and update any in-text citations to match accordingly. Please see our Supporting Information guidelines for more information: http://journals.plos.org/plosone/s/supporting-information.

Reviewers' comments:

Reviewer's Responses to Questions

**Comments to the Author**

1. Is the manuscript technically sound, and do the data support the conclusions?

Reviewer #1: Yes

Reviewer #2: Yes

2. Has the statistical analysis been performed appropriately and rigorously? 

Reviewer #1: I Don't Know

Reviewer #2: Yes

3. Have the authors made all data underlying the findings in their manuscript fully available?

Reviewer #1: Yes

Reviewer #2: Yes

4. Is the manuscript presented in an intelligible fashion and written in standard English?

Reviewer #1: Yes

Reviewer #2: Yes

5. Review Comments to the Author

Reviewer #1: Title: Attentional and Executive Functions in Children and Adolescents with Developmental Coordination Disorder and the Influence of Comorbid Disorders: A Systematic Review of the Literature.

The research justification and aim are clear and relevant; the article is good and useful to improve the future intervention. The article seems solid and the language helps keep track. No important or relevant references appear in the abstract, it has to improve.

- It must be verified that the set of Citations used in the text are correctly cited in the References section (and vice versa) and that, in each case, the DOI or URL corresponding to each cited work is located. The references inserted in the text must also be verified according to APA regulations.

INTRODUCTION. The theoretical review that begins in the introduction of the article reflects a deep exploration but it would be interesting to use recent studies (last 5 years).

Perhaps, expressions such as "most recent" should be reviewed if there are other subsequent references later. Regarding what is already known about the subject of the article, it is recommended to specify why it is an important study and how it differs from previous publications.

The research question should be added clearly in the introduction. Also, the objective that appears in the summary does not match (or is misleading) with what appears as a research question in the objectives. It would be interesting to improve this aspect.

Methods. The methodology is appropriate in terms of the selection criteria. The variables involved are clearly defined. They should review the description of the content in table 1, since what appears in table 1 partially coincides with what is discussed in the methodology section. There are enough details to replicate the study but it would be interesting to add the two or three elements that the NOS includes in each domain and why it was not suitable for the present study.

Results. The results are arranged by categories, which makes it easier to read and understand. The text is not repetitive with respect to the data in the table.

Statistically significant results are clear in some categories, it is recommended to do this in all categories.

In the sustained attention category, the relationship between DCD and ADHD is sometimes confusing. It would be convenient to clarify this.

Although the data is presented clearly and the table format is correct, the table is presented in alphabetical order according to authors, the results are arranged according to categories.

The quality items should be specified in table 2 to facilitate reading.

Discussion and conclusions. It is recommended to relate the content of the discussion with what was previously stated in the introduction and avoid explaining future prospects in this section.

Future research adds that future studies should also focus on children under 8 and over 12 years old, why?

Reviewer #2: The present work shows the results obtained from a systematic review of the literature whose main objective is to identify the attentional and executive deficits present in children with developmental coordination disorder (DCD), for which the type of cognitive tasks, the modality of the tasks (verbal and non-verbal) and the influence of comorbid disorders on the attentional and executive profiles were systematically considered.

The presented results contribute to a better understanding of the cognitive profile associated with CDD, revealing as one of the main findings to be considered that executive impairments are common in CDD (especially in certain areas). Therefore, one of the main implications to be taken into account, and which emerges from the review conducted in this paper, is the consideration of possible attentional and executive impairments in CDD children as inherent to the disorder itself, which, in itself, is one of the strengths of this review.

Another of the main strengths of this review is at the methodological level, as it is a comprehensive review that includes a wide variety of research terms and different types of experimental tasks, as well as standardized neuropsychological tests for the selection of studies. Another strength is the consideration of studies using samples of DCD children with comorbid disorders.

A possible limitation arising from the above consideration is that at certain moments and, assuming the importance of presenting the information in the exhaustive way in which it is done, the reading of the document becomes certainly cumbersome and complex. It would be desirable to review this issue.

Other possible limitations that may be found in this work are assumed by the authors themselves in the section described for this purpose, such as the creation of their own quality assessment tool or the inclusion of studies that, basically due to the number of samples, cannot be considered relative. In this case, the authors warn the reader to take the conclusions drawn with sufficient caution. Furthermore, in the section on Directions for Future Research, lines of research are suggested that also respond to another series of limitations that can be found in the review carried out, such as including studies of self-regulation, extending the age of the sample, verifying whether the executive deficit is prolonged or whether it is a developmental delay, or verifying the overlap between executive functions and motor skills. In this sense, it is desirable that the conclusions section appears before Strenghts and Limitations and Directions for Future Research.

6. PLOS authors have the option to publish the peer review history of their article (what does this mean?). If published, this will include your full peer review and any attached files.

Reviewer #1: **Yes: **Zaira Santana Amador

Reviewer #2: No

---

## [Author Response · Author response to Decision Letter 0]

25 Mar 2021

The authors have revised the comments in the Decision Letter and have changed the manuscript accordingly. The "Response to Reviewers" document contains all the answers/changes that the authors have made.

---

## [Decision Letter · Decision Letter 1]

10 May 2021

Attentional and Executive Functions in Children and Adolescents with Developmental Coordination Disorder and the Influence of Comorbid Disorders: A Systematic Review of the Literature

PONE-D-20-33814R1

Dear Dr. Lippé,

We’re pleased to inform you that your manuscript has been judged scientifically suitable for publication and will be formally accepted for publication once it meets all outstanding technical requirements.

Kind regards,

Celestino Rodríguez, PhD

Academic Editor

PLOS ONE

Additional Editor Comments (optional):

Reviewers' comments:

Reviewer's Responses to Questions

**Comments to the Author**

1. If the authors have adequately addressed your comments raised in a previous round of review and you feel that this manuscript is now acceptable for publication, you may indicate that here to bypass the “Comments to the Author” section, enter your conflict of interest statement in the “Confidential to Editor” section, and submit your "Accept" recommendation.

Reviewer #1: All comments have been addressed

Reviewer #2: All comments have been addressed

2. Is the manuscript technically sound, and do the data support the conclusions?

Reviewer #1: Yes

Reviewer #2: Yes

3. Has the statistical analysis been performed appropriately and rigorously? 

Reviewer #1: Yes

Reviewer #2: Yes

4. Have the authors made all data underlying the findings in their manuscript fully available?

Reviewer #1: Yes

Reviewer #2: Yes

5. Is the manuscript presented in an intelligible fashion and written in standard English?

Reviewer #1: Yes

Reviewer #2: Yes

6. Review Comments to the Author

Reviewer #1: (No Response)

Reviewer #2: Most of the changes initially proposed to the authors of the manuscript have been implemented, although it is true that some of them have not been implemented favourably.

I believe that it was not necessary to change the name of the Objectives section to State of the Knowledge. The only request was that the objective that should appear in the Introduction section should be clearer and, above all, consistent with what was stated in the initial summary.

The clarification included in the Methodology section makes it easier to understand the content of table 1.

Similarly, the changes made in the Results section make the results clearer to understand. Even so, it is still a complicated section.

Finally, the changes in the presentation of the Conclusions, Strengths and Limitations of the current review and Directions for Future Research make the manuscript more consistent and coherent.

7. PLOS authors have the option to publish the peer review history of their article (what does this mean?). If published, this will include your full peer review and any attached files.

Reviewer #1: **Yes: **Dra. Zaira Santana Amador

Reviewer #2: No

---

## [Editor Report · Acceptance letter]

20 May 2021

PONE-D-20-33814R1 

Attentional and executive functions in children and adolescents with developmental coordination disorder and the influence of comorbid disorders: A systematic review of the literature 

Dear Dr. Lippé:

I'm pleased to inform you that your manuscript has been deemed suitable for publication in PLOS ONE. Congratulations! Your manuscript is now with our production department. 

Kind regards, 

on behalf of

Dr. Celestino Rodríguez 

Academic Editor

PLOS ONE